# ARHGAP45 controls naïve T- and B-cell entry into lymph nodes and T-cell progenitor thymus seeding

Le He[1,2], Marie-Pierre Valignat[3], Lichen Zhang[2], Lena Gelard[1,4], Fanghui Zhang[1,2], Valentin Le Guen[1], Stéphane Audebert[5], Luc Camoin[5], Even Fossum[6], Bjarne Bogen[6], Hui Wang[2], Sandrine Henri[1], Romain Roncagalli[1], Olivier Theodoly[3], Yinming Liang[2,*], Marie Malissen[1,4,7,**] & Bernard Malissen[1,4,7,***]

## Abstract

T and B cells continually recirculate between blood and secondary lymphoid organs. To promote their trans-endothelial migration (TEM), chemokine receptors control the activity of RHO family small GTPases in part via GTPase-activating proteins (GAPs). T and B cells express several RHO-GAPs, the function of most of which remains unknown. The ARHGAP45 GAP is predominantly expressed in hematopoietic cells. To define its *in vivo* function, we describe two mouse models where ARHGAP45 is ablated systemically or selectively in T cells. We combine their analysis with affinity purification coupled to mass spectrometry to determine the ARHGAP45 interactome in T cells and with time-lapse and reflection interference contrast microscopy to assess the role of ARGHAP45 in T-cell polarization and motility. We demonstrate that ARHGAP45 regulates naïve T-cell deformability and motility. Under physiological conditions, ARHGAP45 controls the entry of naïve T and B cells into lymph nodes whereas under competitive repopulation it further regulates hematopoietic progenitor cell engraftment in the bone marrow, and T-cell progenitor thymus seeding. Therefore, the ARGHAP45 GAP controls multiple key steps in the life of T and B cells.

**Keywords** ARHGAP45; cell migration; chemotaxis; GTPase-activating protein; lymphocyte
**Subject Categories** Immunology; Signal Transduction; Stem Cells & Regenerative Medicine

## Introduction

Secondary lymphoid organs are anatomical sites in which adaptive immune responses are initiated. They include lymph nodes (LNs), and the spleen and naïve T cells continually recirculate between them and the blood. Entry into LNs involves a multistep cascade in which the lymphocyte-homing receptor L-selectin (CD62L) supports initial rolling of blood-borne, naive T cells along high endothelial venules (HEVs). The chemokines CCL19 and CCL21 on the luminal surface of HEVs bind to the CCR7 chemokine receptor expressed on rolling naïve T cells, leading successively to LFA-1-ICAM-1-mediated firm adhesion, T-cell polarization, and subsequent crawling over and diapedesis through HEVs. Blood-borne, naive B cells also use CCR7 together with the CXCR4 and CXCR5 chemokine receptors to enter LNs (Schulz *et al*, 2016).

Among the eight sub-families that constitute the RHO family of small GTPases, the RAC, RHO and CDC42 sub-families control cell polarity, shape, and migration by regulating actin cytoskeletal dynamics (Lawson & Ridley, 2018). The activity of these small GTPases is controlled by guanine nucleotide exchange factors (GEFs) and GTPase-activating proteins (GAPs) that allow them to cycle between a GTP-bound form which activates downstream effectors and an inactive GDP-bound form. To promote trans-endothelial migration (TEM), CCR7 activates various phospholipase C (PLC) and phosphoinositide 3-kinase (PI3K) isoforms. PLCs generate inositol trisphosphate and diacylglycerol (DAG), whereas PI3Ks generate phosphatidylinositol 3,4,5 trisphosphate (PIP3) (Schulz *et al*, 2016). Such intracellular second messengers control the intracellular distribution and function of RHO GEFs and GAPs.

T cells express more than 20 GAPs, and the function and mechanism of action of most of them remain to be elucidated (Stein & Ruef, 2019). We recently identified the ARHGAP45 GAP among the

---

1 Centre d'Immunologie de Marseille-Luminy, INSERM, CNRS, Aix Marseille Université, Marseille, France
2 Henan Key Laboratory for Immunology and Targeted Therapy, School of Laboratory Medicine, Xinxiang Medical University, Xinxiang City, China
3 LAI, CNRS, INSERM, Aix Marseille Univ, Marseille, France
4 Centre d'Immunophénomique, INSERM, CNRS UMR, Aix Marseille Université, Marseille, France
5 CNRS, INSERM, Institut Paoli-Calmettes, CRCM, Marseille Protéomique, Aix Marseille Univ, Marseille, France
6 Institute of Clinical Medicine, University of Oslo and Oslo University Hospital, Oslo, Norway
7 Laboratory of Immunophenomics, School of Laboratory Medicine, Xinxiang Medical University, Xinxiang City, China
 *Corresponding author. Tel: +86 373 383 1237; E-mail: yinming.liang@foxmail.com
 **Corresponding author. Tel: +33 6 33 24 54 04; E-mail: malissen@ciml.univ-mrs.fr
 ***Corresponding author. Tel: +33 7 86 28 29 83; E-mail: bernardm@ciml.univ-mrs.fr

signaling protein complexes that assemble in mouse primary T cells (Voisinne *et al*, 2019; Mori *et al*, 2021). ARHGAP45, also known as HMHA1 (human minor histocompatibility antigen 1), has been first described in human and comprises an N-terminal BAR domain followed by a C1 and a RHO GAP domain (de Kreuk *et al*, 2013). BAR domains consist of a helical bundle of 200–280 amino acids that associates in antiparallel fashion to form dimers that bind to membranes according to their curvature (Carman & Dominguez, 2018), whereas C1 domains bind membrane-bound DAG. Heterologous expression of human ARHGAP45 in HeLa cells showed that it regulates the actin cytoskeleton and cell spreading through undefined mechanism (de Kreuk *et al*, 2013). ARHGAP45 GAP activity is autoinhibited by an intramolecular interaction involving the BAR domain (de Kreuk *et al*, 2013), suggesting that ARHGAP45 adopts an active conformation when its C1 and BAR domains simultaneously bind to DAG-containing membranes with an appropriate curvature. To analyze the role of ARHGAP45 *in vivo*, we developed two mouse models where ARHGAP45 is ablated either systemically or selectively in T cells. We also determined the constellation of proteins that specifically interact with ARHGAP45 in T cells and assessed the role of ARGHAP45 in T-cell deformability, polarization and motility. Altogether, our data demonstrate that ARHGAP45 plays a critical role in the entry of naïve T and B cells into LNs, in the engraftment of hematopoietic progenitor cells in the bone marrow (BM), and in T-cell progenitor thymus seeding.

## Results

### Characterization of the ARGHAP45 interactome in Jurkat T cells

Although ARHGAP45 is predominantly expressed in hematopoietic cells (www.immgen.org, http://biogps.org, https://genevisible.com/search), previous studies on ARHGAP45 relied on overexpression in heterologous cells (de Kreuk *et al*, 2013). Therefore, before analyzing the functional consequences resulting from the lack of ARHGAP45 *in vivo*, we used quantitative interactomics (Voisinne *et al*, 2019) to determine which proteins associate with endogenous ARHGAP45 molecules in T cells. Using CRISPR/Cas9 editing, the ARHGAP45 molecules present in the human leukemic T-cell line Jurkat were tagged at their amino terminus with an affinity Twin-Strep-tag (OST; Fig 1A). The proteins ("the preys") associating to the ARHGAP45$^{OST}$ "bait" were affinity purified before or after stimulation with anti-CD3 for 120 s and analyzed by MS. To distinguish proteins truly associating with ARHGAP45 from nonspecific contaminants, we compared our data with control AP-MS experiments involving wild-type (WT) Jurkat T cells. ARHGAP45 interactors were identified (Fig 1B and Dataset EV1), and the ARHGAP45-prey stoichiometry was also measured using intensity-based absolute quantification (Voisinne *et al*, 2019). By determining the number of copies per cell of each protein expressed in Jurkat cells (Dataset EV2) and combining them with interaction stoichiometries, we organized the proteins interacting with ARHGAP45 (the "ARHGAP45 interactome") into a stoichiometry plot (Fig 1C).

Analysis of the GAP activity of ARHGAP45 in a cell-free system showed that it can catalyze RHOA, RAC1 and CDCD42 GTP hydrolysis (de Kreuk *et al*, 2013). In contrast, our stoichiometry plot showed that RHOA was the major RHO GTPase capable of

associating with ARHGAP45 *in cellulo* and constituted one of the most abundant ARHGAP45 interactors (Fig 1C), suggesting that in T cells ARHGAP45 primarily regulates the RHOA GTPase. RHOC, a second member of the RHO family of small GTPases, was also found among ARHGAP45 interactors, and its interaction stoichiometry was 31-fold lower than that of the ARHGAP45-RHOA interaction (Fig 1C). This might result from the fact that ARHGAP45 is a better GAP for RHOC than RHOA, leading to its rapid dissociation from ARHGAP45 after GTP hydrolysis. Some BAR domain-containing proteins bind to members of the 14-3-3 protein family in a phospho-serine-dependent manner (Carman & Dominguez, 2018). Along that line, YWHAQ, also known as 14-3-3 protein θ, associated with ARHGAP45 in a TCR-inducible manner (Fig 1C), a finding consistent with the presence of several TCR-inducible phosphorylated serine residues in ARGHAP45 (Locard-Paulet *et al*, 2020). Proteins involved in regulation of cortical actin tension (MYO1G), cell migration (CORO1C), vesicular transport (RAB11B), or colocalizing with the chemokine receptor CXCR4 and F-actin in T cells (DBN1) also associated with ARHGAP45, however, with a lower stoichiometry than RHOA and YWHAQ (Fig 1C). Interestingly, ARHGAP45 associated with another BAR-containing GAP protein called GMIP (also known as ARHGAP46; Fig 1C), a result consistent with the possibility for distinct RHO GAP to heterodimerize via their BAR domain (Carman & Dominguez, 2018). Therefore, the composition of the ARHGAP45 interactome of Jurkat T cells suggests that ARHGAP45 acts as a GAP specific for RHOA and presumably RHOC.

### Effect of ARHGAP45 deficiency on T- and B-cell development

To analyze the role of ARHGAP45 *in vivo*, we established mice homozygous for an *Arhgap45* allele lacking a critical exon (*Arhgap45* exon 4; Fig EV1A). The resulting *Arhgap45$^{-/-}$* mice were born at expected Mendelian frequencies and lacked detectable ARHGAP45 protein as exemplified using developing T cells and mature T and B cells (Fig EV1B). *Arghap45$^{-/-}$* thymi were of normal size (Fig 2A) and contained normal numbers of DN2, DN3, DN4, and DP cells (Fig 2B; see legend for a definition of T-cell developmental stages). The presence of reduced numbers of DN1 cells (1.5-fold), and of both CD4$^+$ (1.3-fold) and CD8$^+$ (1.4-fold) SP thymocytes suggested that thymus seeding and transit from the medulla to the cortex, two steps depending on CCR7 signals (Kurobe *et al*, 2006; Calderon & Boehm, 2011), were slightly impeded by the lack of ARHGAP45. The presence of normal DN2-3 cell numbers further suggested that some compensatory cell divisions occurred at the DN1 to DN2-3 transition.

The numbers of T and B cells found in the blood of *Arhgap45$^{-/-}$* mice were 3.7- and 3.2-fold reduced, respectively (Fig 2C). *Arhgap45$^{-/-}$* LNs had a reduced cellularity (Fig 2A) due to diminished numbers of naïve T (2.2-fold) and B (2.5-fold) cells (Fig 2D), whereas *Arhgap45$^{-/-}$* Peyer's patches showed an even stronger reduction of naïve T (4-fold) and B (17-fold) cell numbers (Fig 2E). In contrast, *Arhgap45$^{-/-}$* spleens showed a normal cellularity with normal T- and B-cell numbers and an almost normal representation of T1, T2, and follicular B cells (Fig 2A and F, see legend for a definition of B-cell developmental stages). Comparable number of effector memory CD4$^+$ and CD8$^+$ T cells were found in *Arhgap45$^{-/-}$* and WT LNs (Fig 2G). Due to the reduced numbers of naïve T cells found in *Arhgap45$^{-/-}$* LNs, increased percentages of effector

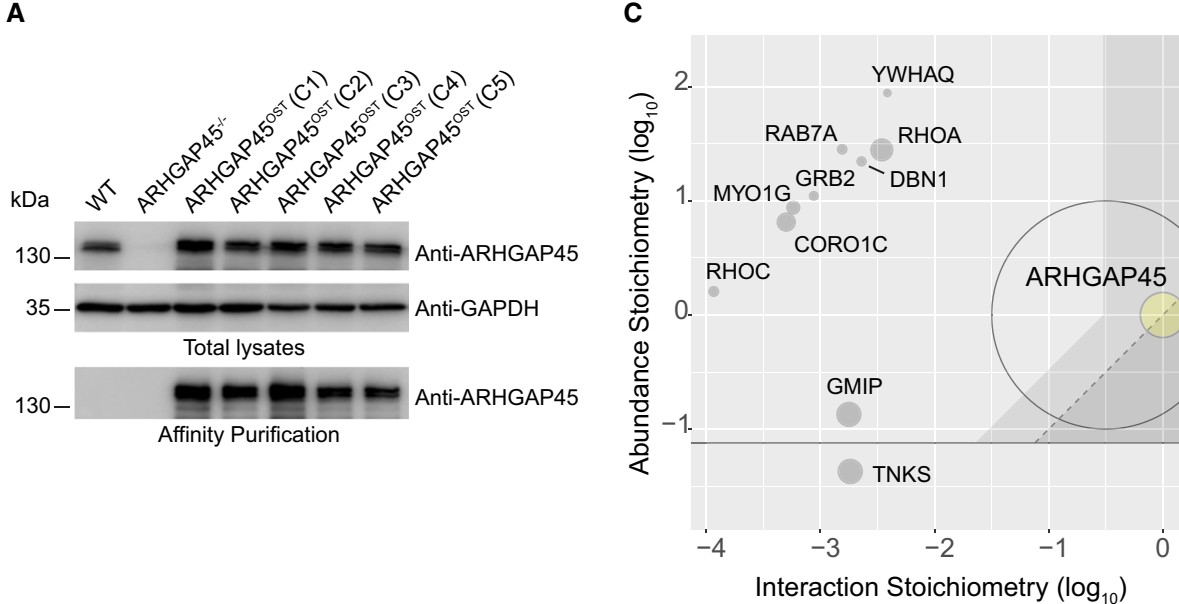

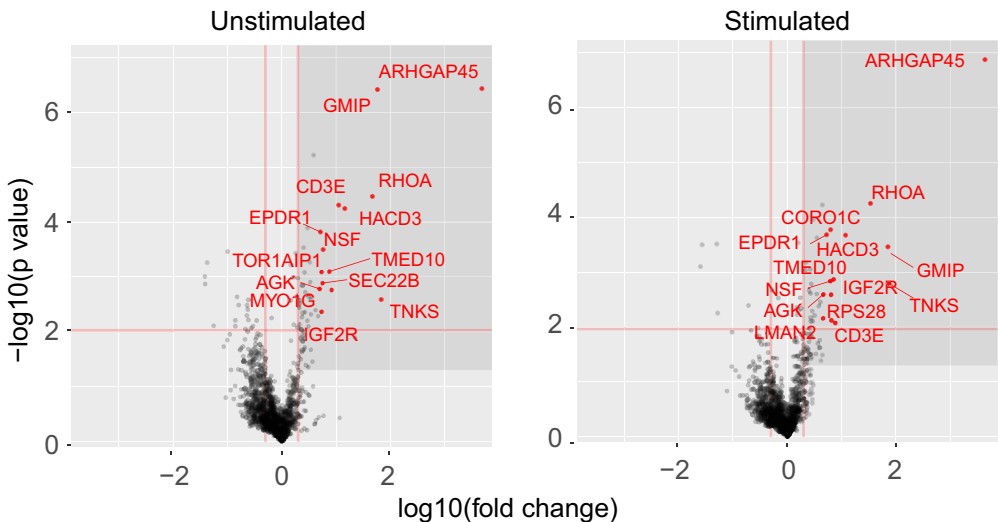

**Figure 1. The ARHGAP45 interactome of Jurkat T cells.**

A  Jurkat T cells (WT), ARHGAP45-deficient Jurkat T cells (ARHGAP45$^{-/-}$), and five independent clones (C1–C5) of Jurkat T cells expressing ARHGAP45$^{OST}$ molecules were analyzed for ARHGAP45 protein expression. Immunoblot analysis of equal amounts of proteins from cell lysates that were either directly analyzed (Total lysates), or subjected to affinity purification on Strep-Tactin Sepharose beads followed by elution of proteins with D-biotin (Affinity purification), and probed with antibody to ARGHAP45 or anti-HSP60 (loading control). Left margin, molecular size in kilodaltons (kDa).

B  Volcano plot showing proteins enrichment (fold change in log$_{10}$ scale) after affinity purification in Jurkat T cells expressing ARHGAP45$^{OST}$ molecules compared to affinity purification in control Jurkat T cells expressing similar levels of WT (untagged) ARHGAP45 proteins prior to (unstimulated) and at 120 s after (stimulated) TCR stimulation. ARHGAP45 interacting proteins with a > 2-fold enrichment and a *P*-value < 0.01 were selected as specific ARHGAP45 interactors (Dataset EV1) and some of them are specified in red. Red lines represent the thresholds set on P-value and enrichment to identify specific ARHGAP45 interactors.

C  ARHGAP45-specific interactors highlighted in the text are displayed in a stoichiometry plot (Voisinne *et al*, 2019) where the ratios of the prey to bait cellular abundances ("abundance stoichiometry") are plotted as a function of their maximal interaction stoichiometries ("interaction stoichiometry"), both using log$_{10}$ scale (see Datasets EV1 and EV2). The ARHGAP45 bait is shown as a yellow dot. The size of the dots is commensurate to the maximal protein enrichment in ARHGAP45$^{OST}$ samples as compared to WT control samples. The TNKS prey for which it was not possible to determine the cellular abundance is shown at the bottom of the stoichiometry plot. For each time point, three independent biological replicates were performed and each biological replicate was analyzed in triplicate by MS.

Source data are available online for this figure.

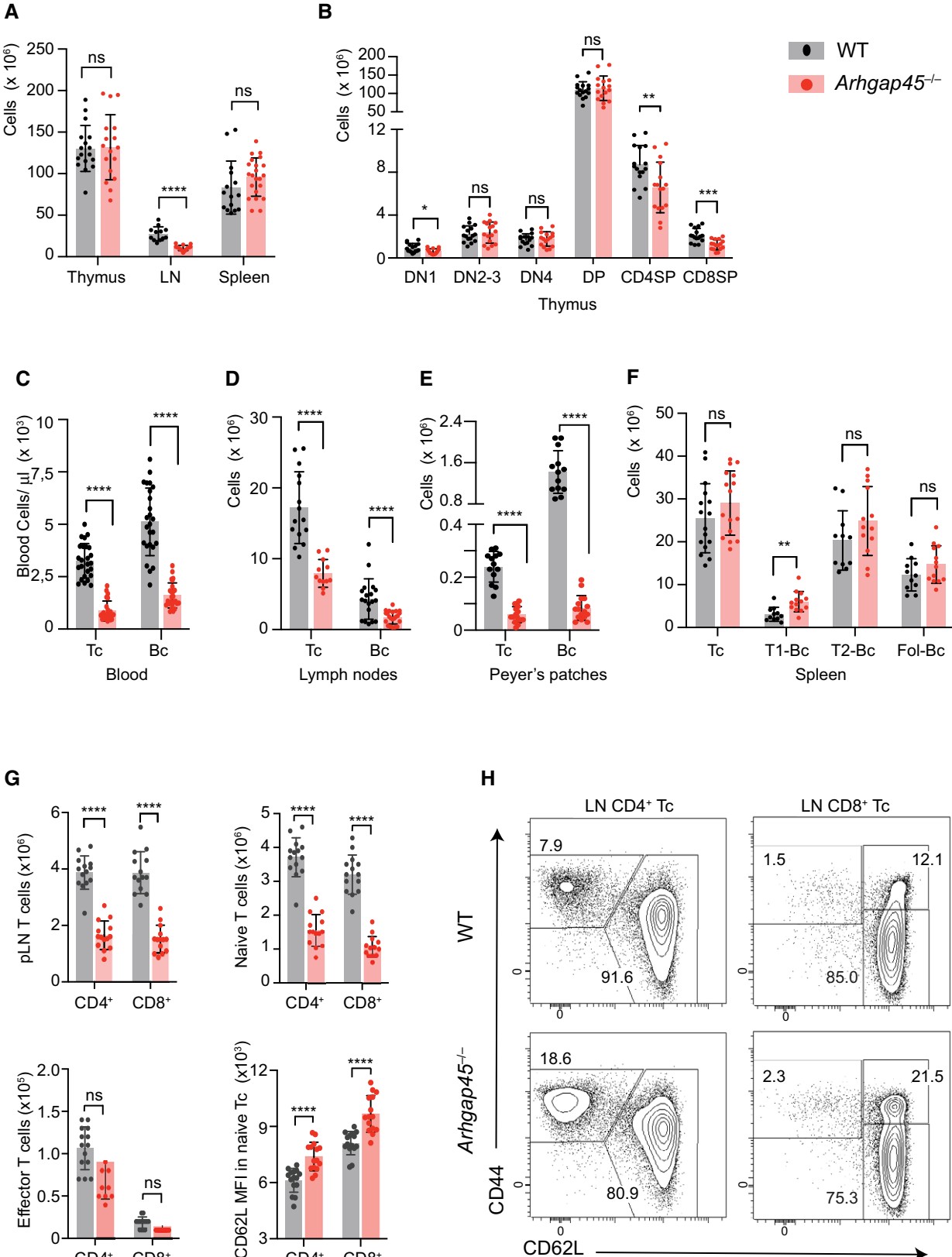

**Figure 2.**

**Figure 2. Development of T and B cells in *Arhgap45*⁻/⁻ mice.**

A  Total cellularity of thymus spleen and of mesenteric and peripheral LNs of WT and *Arhgap45*⁻/⁻ mice (see key in upper right corner).

B  Upon thymus colonization, ETP develop into CD4⁻CD8⁻double negative (DN) cells that mature into CD4⁺CD8⁺ double positive (DP) cells, some of which proceed further into CD4⁺ and CD8⁺ single positive (SP) cells that egress from the thymus. Based on the expression of CD25 and CD44, DN cells can be further organized according to the following developmental series: DN1 (CD44⁺CD25⁻) → DN2 (CD44⁺CD25⁺) → DN3 (CD44⁻CD251⁺) → DN4 (CD44⁻CD25⁻). After excluding cells positive for CD11b, CD11c, CD45R, or CD161c, WT and *Arhgap45*⁻/⁻ thymocytes were analyzed by flow cytometry for expression of CD4, CD8, CD25, and CD44 and the numbers of cells present in each of the specified T-cell developmental stages determined.

C  Numbers of T and B cells found in the blood of WT and *Arhgap45*⁻/⁻ mice.

D  Numbers of T and B cells found in pooled mesenteric and peripheral LNs of WT and *Arhgap45*⁻/⁻ mice.

E  Numbers of T and B cells found in the Peyer's patches of WT and *Arhgap45*⁻/⁻ mice.

F  In the spleen, IgM$^{hi}$IgD$^{lo}$ transitional 1 (T1-Bc) B cells constitute recent immigrant from the BM that develop into IgM$^{hi}$IgD$^{hi}$ transitional (T2-Bc) B cells, which differentiate into mature IgM$^{lo}$IgD$^{hi}$, or follicular recirculating B cells Fol-Bc (Carsetti, 2004)). WT and *Arhgap45*⁻/⁻ splenocytes were analyzed by flow cytometry for expression of CD19, CD45R, IgM and IgD and the numbers of cells present in each of the specified B-cell developmental stages determined.

G  Numbers of total CD4⁺ and CD8⁺ T cells, and of naïve and effectors CD4⁺ and CD8⁺ T cells found in peripheral LNs of WT and *Arhgap45*⁻/⁻ mice. Also shown are CD62L levels (MFI) on naïve CD4⁺ and CD8⁺ T cells from WT and *Arhgap45*⁻/⁻.

H  CD4⁺ and CD8⁺ T cells from peripheral LNs analyzed for expression of CD44 and CD62L. Numbers in quadrants indicate percent naïve (CD44$^{lo}$CD62L$^{hi}$) and central memory CD8⁺ T cells (CD44$^{hi}$CD62L$^{hi}$).

Data information: In (A)–(H) the results for each mouse are shown as a dot and correspond to three to four experiments involving a total of 12–25 mice. *$P \leq 0.033$, **$P \leq 0.002$, ***$P \leq 0.001$, ****$P \leq 0.0001$; unpaired Student's *t*-test. Mean and SD (A, B, G) or SEM (C, D, E, F) are also shown.

memory cells were observed in *Arhgap45*⁻/⁻ CD44-CD62L dot plots (Fig 2H). Therefore, considering that the spleen differs from LNs and Peyer's patches by its lacks of HEV (Schulz *et al*, 2016), and that effector memory T cells primarily reach LNs via afferent lymphatic vessels (Jackson, 2019), our data suggest that the absence of ARHGAP45 specifically impedes the TEM of naïve T and B cells through the HEV of LNs and Peyer's patches.

### ARHGAP45 deficiency impedes LN entry rate of naïve T and B cells

To determine whether the reduced numbers of naïve T and B cells found in the LNs of *Arhgap45*⁻/⁻ mice were due to defective LN entry rate, we measured the short-term LN homing efficiency of adoptively transferred ARHGAP45-deficient T and B cells and compared it to that of co-transferred WT T and B cells. Accordingly, total spleen cells from WT and *Arhgap45*⁻/⁻ mice were labeled with CTV and CMTPX dyes, respectively, mixed at a 1 to 1 ratio, and injected intravenously into WT recipient mouse. For both T and B cells, the ratio of *Arhgap45*⁻/⁻ to WT cells that entered LNs and spleen was measured 4 h after transfer (Fig 3A). Significantly decreased *Arhgap45*⁻/⁻/WT ratios were observed in LNs for both T (1.8-fold) and B (2.2-fold) cells as compared to pre-injection ratios (Fig 3B). In contrast, ARHGAP45 deficiency was without measurable effects on the entry of B and T cells into the spleen (Fig 3B). In control experiments involving 1 to 1 mixture of WT spleen cells labeled with CTV or CMTPX, no decrease was observed in the T- and B-cell CTV-CMTPX ratio found in LNs (Fig 3A and B). The naïve T cells found in the blood and LNs of WT and *Arhgap45*⁻/⁻ mice expressed levels of CD62L (Fig 2H), CCR7 and LFA-1 (Fig. 3C and D) comparable or even slightly increased (CD62L) as compared to their WT counterparts. Therefore, naïve T and B cells deficient in ARHGAP45 showed an impeded entry into LNs despite their expression of normal levels of CD62L, CCR7 and LFA-1 molecules.

### Lack of ARHGAP45 in T cells suffices to account for their defective LN entry

Our analysis of LN entry rate via short-term T- and B-cell transfer suggested that the diminished entry into LNs observed in

*Arhgap45*⁻/⁻ mice is intrinsic to T and B cells and not due to indirect effects resulting from the lack of migratory dendritic cells (DC) which are known to modulate the function of HEV (Girard *et al*, 2012) or from a role of ARHGAP45 in HEV. To directly assess the effect resulting from the deletion of ARHGAP45 in and only in T cells, we introduced a *Cd4*-Cre transgene onto mice homozygous for a floxed *Arghap45* allele (Fig. EV1C and Materials and Methods). Due to the selective deletion of the *Arghap45* gene from the DP stage onwards, ARHGAP45 expression was retained in follicular B cells but not in naive T cells (Fig EV1B), and the cellularity of *Arhgap45*$^{TΔ/ΔT}$ thymi was not affected (Fig EV2A). The specific absence of ARHGAP45 expression in *Arhgap45*$^{ΔT/ΔT}$ naïve T cells reduced their numbers in the blood and LNs with a magnitude comparable to that observed in *Arhgap45*⁻/⁻ mice (Fig EV2B–E). In contrast, it had no effect on the absolute numbers of *Arhgap45*$^{ΔT/ΔT}$ B cells found in the blood and LNs. Akin to the spleen of *Arhgap45*⁻/⁻ mice, *Arhgap45*$^{ΔT/ΔT}$ spleen contained normal numbers of T and B cells (Fig EV2F–G). Therefore, T-cell-specific inactivation of ARHGAP45 suffices to reduce the number of T cells present in the blood and LNs.

### Impaired chemokine-driven migration of *Arhgap45*⁻/⁻ naïve T and B cells

The impact of ARHGAP45 deficiency on chemokine-induced T- and B-cell migration was analyzed next using Transwell migration assays. When compared to their WT counterparts, *Arhgap45*⁻/⁻ CD4⁺ and CD8⁺ naïve T cells showed markedly diminished migration in response to the CCL19 (CD4⁺: 7.6-fold less, CD8⁺: 3.6-fold less), CCL21 (CD4⁺: 10-fold less, CD8⁺: 8.5-fold less) and CXCL12 (CD4⁺: 5.5-fold less, CD8⁺: 4.6-fold less) chemokines (Fig 4A and B). Likewise, the migration of *Arhgap45*⁻/⁻ naïve B cells in responses to those chemokines was also affected (CCL19: 2.4-fold less, CCL21: 2.2-fold less, and CXCL12: 2.0-fold less; Fig 4C). Similar results were obtained when Transwell membrane inserts were precoated with ICAM-1 (Fig 4D). To distinguish random migratory behavior in response to chemokine (chemokinesis) from directional migration toward a chemokine gradient (chemotaxis), CCL19 was placed in both the upper and lower Transwell chambers. No WT T-

cell migration occurred under that condition, confirming that our assay measured chemotaxis (Fig 4D). Consistent with the expression of ARHGAP45 in B cells but not in T cells of $Arhgap45^{\Delta T/\Delta T}$ mice,

only the former retained the capacity to properly migrate in response to CCL19 (Fig 4E). Following *in vitro* stimulation with anti-CD3 and anti-CD28, $Arhgap45^{-/-}$ activated T cells were capable of normal

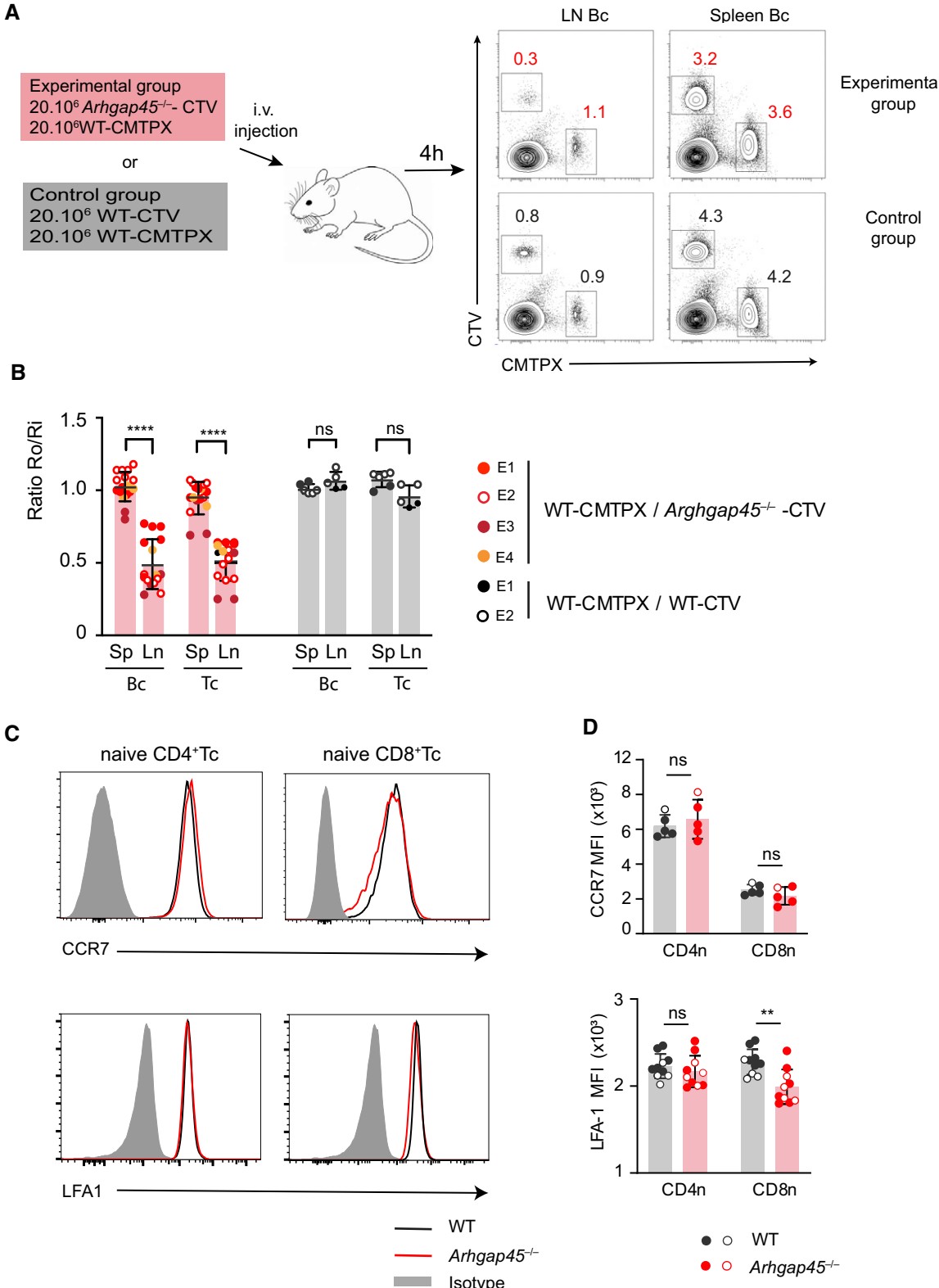

Figure 3.

**Figure 3.  *Arhgap45*<sup>−/−</sup> naïve T and B cells showed impeded LN entry.**

A  CTV-labeled splenocytes (20 × 10⁶ cells) from either *Arhgap45*⁻/⁻ or WT mice were mixed with CMTPX-labeled WT splenocytes (20 × 10⁶) and injected intravenously in WT recipient mice. After 4 h, single-cell suspensions were prepared from the spleen and mesenteric LNs (mLN) and the percentages of CTV- and CMTPX-labeled B and T cells determined by flow cytometry.
B  The ratio of CTV/CMTPX-labeled cells present in each organ (Ro) was determined and normalized by dividing it with Ri, the ratio of CTV/CMTPX-labeled cells present in the cell mixtures prior to injection. Ro/Ri ratio corresponding to each of the specified mice are shown. Data shown for *Arhgap45*⁻/⁻ mice correspond to 4 independent experiments (E1–E4) involving a total of 15 individual mice whereas data shown for WT mice correspond to two independent experiments (E1 and E2) involving a total of five individual mice. The *Arhgap45*⁻/⁻ mice analyzed on the same day as the control mice corresponded to the E1 and E2 experiments. Mean and SD are shown. ns, non-significant, ****$P \leq 0.0001$; multiple unpaired *T*-tests.
C  Expression of CCR7, and LFA-1 on naïve CD4⁺ and CD8⁺ T cells found in LNs of WT and *Arhgap45*⁻/⁻ mice, analyzed by flow cytometry. Gray shaded curves, isotype control staining. D Quantification of data shown in (C). The MFI corresponding to each analyzed mouse is shown together with mean and SD. Each dot represents an individual mouse and CD4⁺ T cells from at least 4 mice have been analyzed for each condition. Mice analyzed on the same day are represented by similar dots (see key in bottom right corner). ns, non-significant, **$P \leq 0.002$; multiple unpaired *T*-tests.

chemokine-driven migration in response to CCL19 (Fig 4F). Therefore, ARHGAP45 was found specifically required for efficient chemokine-induced migration of naive T and B cells but dispensable for chemokine-induced migration of activated T cells.

### *Arhgap45*<sup>−/−</sup> activated T cells show normal crawling along 2D surface coated with ICAM-1

We used time-lapse video microscopy to investigate the role of ARHGAP45 on T-cell migration and crawling on 2D surface coated with ICAM-1. Activated T cells used in these experiments were stimulated with anti-CD3 and anti-CD28 for 2 days and "rested" in IL-2 for 2 days before examining their migratory capacity. As previously reported (Smith *et al*, 2003; Valignat *et al*, 2013), when placed on ICAM-1-coated surfaces in absence of chemokine, WT activated T cells showed a sustained motility (Fig 5A) and a polarized morphology with a highly dynamic leading edge region and a non-adherent contractile trailing edge region (Movie EV1). *Arhgap45*⁻/⁻ activated T cells were also motile (Fig 5A) and had a polarized morphology similar to that of WT activated T cells (Movie EV2). Moreover, the lack of ARHGAP45 was without measurable effect on the migratory speed of activated T cells (Fig 5B and C). By measuring the percentage of activated T cells remaining adherent on ICAM-1-coated surface after applying a shear stress of 1 dynes/cm² for 120 s, we showed that the lack of ARHGAP45 was without measurable effect on adhesion (Fig 5D). Therefore, *Arhgap45*⁻/⁻ activated T cells showed normal chemokine-independent crawling along ICAM-coated surface, suggesting that ARHGAP45 does not have a generic role in the coordinated regulation of LFA-1 affinity and actomyosin cytoskeleton contractility (Heasman *et al*, 2010).

### Impaired deformability and motility of *Arhgap45*<sup>−/−</sup> naïve T cells in response to CCL21 and CCL19

When placed under static conditions on 2D surface coated with ICAM-1, WT naïve T cells require chemokines such as CCL21 or CCL19 to polarize and subsequently migrate (Fig EV3A and B), a phenomenon known as chemokinesis. Under those conditions, *Arhgap45*⁻/⁻ naïve T cells showed a 2-fold slower motility as compared to WT naïve cells (Fig 5E–G and Movies EV3 and EV4). When subjected to a shear stress of 1 dynes/cm² for 120 s, the number of *Arhgap45*⁻/⁻ naïve T cells remaining adherent to the ICAM-1- and chemokine-coated surface was comparable to that of WT naïve T cells (Fig 5H). Observations at high magnification

allowed to determine cell contours and compare the temporal evolution of the morphology of WT and *Arhgap45*⁻/⁻ naïve T cell migrating under static conditions on surface coated with ICAM-1 and chemokines (Fig 6A and B). It permitted to broke down the trajectory of WT naïve T cells into a series of cycles that comprise each a phase of polarization and rapid migration followed by a phase of arrest where cells become more rounded and less adherent. At the end of a given cycle, naïve T cells embark into a novel cycle that offers them the possibility to migrate into another direction (Movie EV3 and Fig 6C). In contrast to WT naïve T cells, once *Arhgap45*⁻/⁻ naïve T cells succeeded polarizing most of them remained locked in such state over the whole observation period (Movie EV4 and Fig 6D).

To quantify these distinct migratory behaviors, the deformation of each individual cell was evaluated in each time frame by calculating the eccentricity of the ellipse that has the same second-moment as the cell contours. The eccentricity, which is the ratio of the distance between the foci of an ellipse and its major axis length, can take values comprised between 0 and 1; with 0 corresponding to a circle and 1 to a line segment. The histograms of instant eccentricity showed a two-state bimodal distribution for WT naïve T cells and a unimodal distribution for *Arhgap45*⁻/⁻ naïve T cells (Fig 6E and G). It confirms that WT naïve T cells alternate between highly motile/polarized and arrested/non-polarized states, whereas *Arhgap45*⁻/⁻ naïve T cells adopt a persistent polarization and slow motility.

Using reflection interference contrast microscopy (RICM), we imaged the adhesion footprint of T cells and determined the percentage of adherent surface in the cell projected area (Fig EV4A–F). Histograms corresponding to the instant percentage of surface adhesion displayed a bimodal distribution for WT naïve T cells and unimodal distribution for *Arhgap45*⁻/⁻ naïve T cells (Fig EV4G–H). 2D plots of instant cell speed versus instant percentage of adherent surface were then built to assess the relationship between the adherence and the motility (Fig 6F and H). *Arhgap45*⁻/⁻ naïve T cells showed a single population of adherent (projected adherent area > 25%) and slow (instant speed < 15 μm/min) cells (Fig 6H), reflecting their unimodal migration pattern with a stable polarization, adhesion and slow speed. In contrast, WT naïve T cells showed a more complex distribution with 3 populations (Fig 6F). Two populations were hardly motile (instant speed < 5 μm/min) with either low (< 20%) or high (> 40%) projected adherent area and corresponded to round cells with varying degree of spreading. The third widely scattered population comprised highly adherent cells (projected adherent area > 20%) with elevated speed (ranging between 15 and 30 μm/

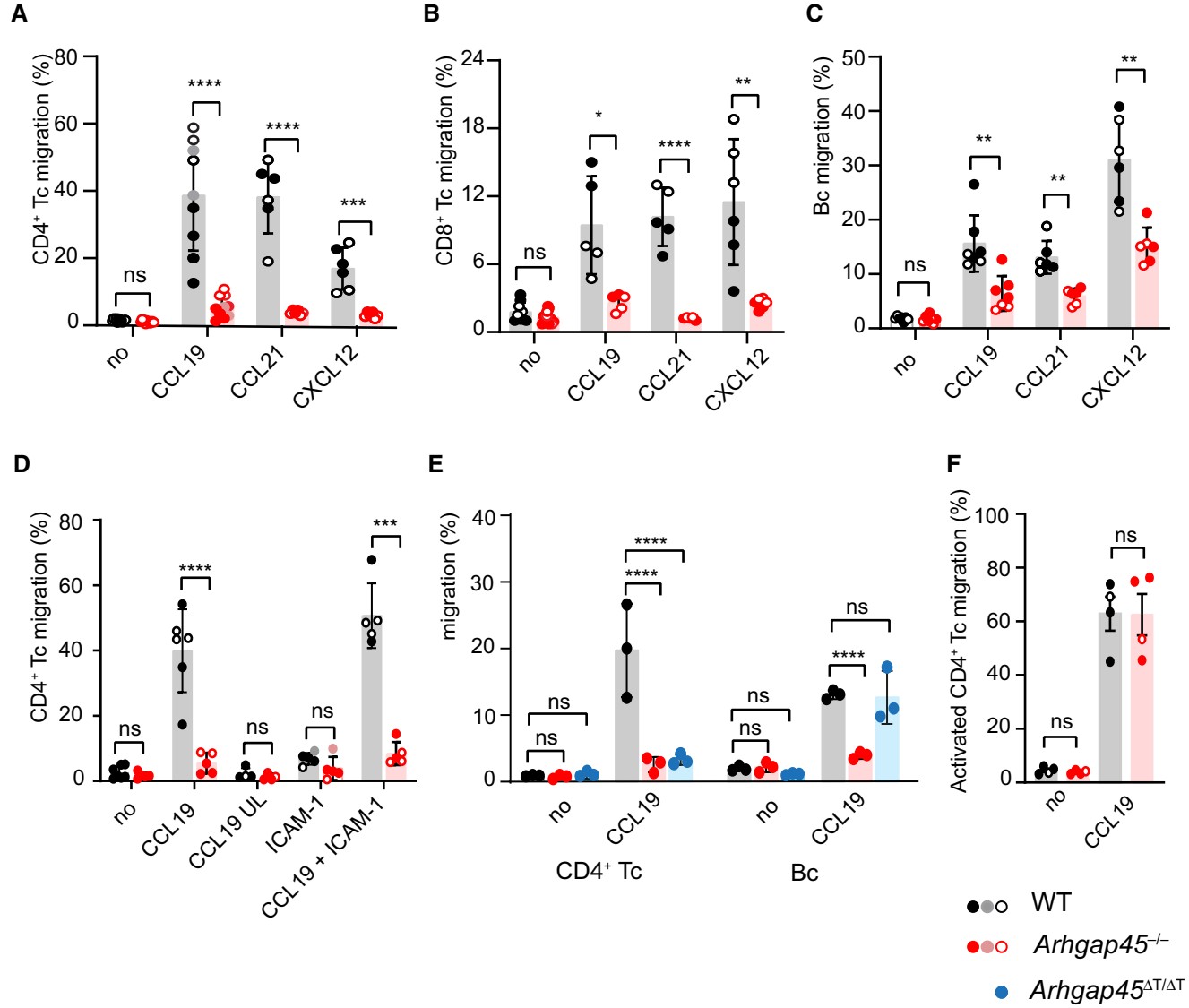

**Figure 4. Reduced chemotaxis of Arhgap45⁻/⁻ naïve T and B cells.**

The specified cells (see lower right corner) were loaded into the upper chamber of a Transwell migration device without (no) or with the specified chemokines in the lower chamber. The percentage of cells that migrated to the bottom chamber after 2 h (T cells) or 4 h (B cells) is shown.

A   Chemotaxis of naïve CD4⁺ T cells from WT and *Arghap45⁻/⁻* mice in response to CCL19, CCL21 and CXCL12.

B   Chemotaxis of naïve CD8⁺ T cells from WT and *Arghap45⁻/⁻* mice in response to CCL19, CCL21 and CXCL12.

C   Chemotaxis of naïve B cells from WT and *Arghap45⁻/⁻* mice in response to CCL19

D   Adding CCL19 in both the upper and lower chambers (CCL19 UL) prevented migration of WT and *Arghap45⁻/⁻* naïve T cells whereas the use of Transwell membrane inserts precoated with ICAM-1 (CCL19 + ICAM-1) did not change the pattern of chemotaxis observed using "bare" membrane inserts (CCL19).

E   Chemotaxis of naïve CD4⁺ T cells and B cells from WT, *Arghap45⁻/⁻*, and *Arghap45^{ΔT/ΔT}* mice in response to CCL19.

F   Chemotaxis of activated CD4⁺ T cells from WT and *Arghap45⁻/⁻* mice in response to CCL19.

Data information: In (A)–(F) each dot represents an individual mouse and as shown CD4⁺ T cells from 3 to 9 mice have been analyzed for each condition. Mice analyzed on the same day are represented by similar dots (see key in bottom right corner). Mean and SD are shown. *P ≤ 0.033, **P ≤ 0.002, ***P ≤ 0.001, ****P ≤ 0.0001; (A, B, C, D, F) multiple unpaired *T*-test; E two-way ANOVA test.

min), corresponding to T cells in a polarized, adherent and rapid migratory state. Therefore, the lack of ARHGAP45 "locked" naïve T cells into an adherent and polarized stiff state that impede their migratory speed. Such poor deformability likely accounts for the diminished migration of *Arhgap45⁻/⁻* naïve T cells through the tight membrane pores of Transwell membranes in response to a gradient of soluble chemokines.

**Lack of ARHGAP45 does not impair *in vivo* T-cell activation by antigen-laden DC**

To assess whether ARHGAP45 also contributed to TCR-dependent T-cell activation in response to antigen-laden antigen presenting cells (APC), mice expressing the OT-I TCR specific for the N4 ovalbumin (OVA) peptide (Barnden *et al*, 1998) were backcrossed onto

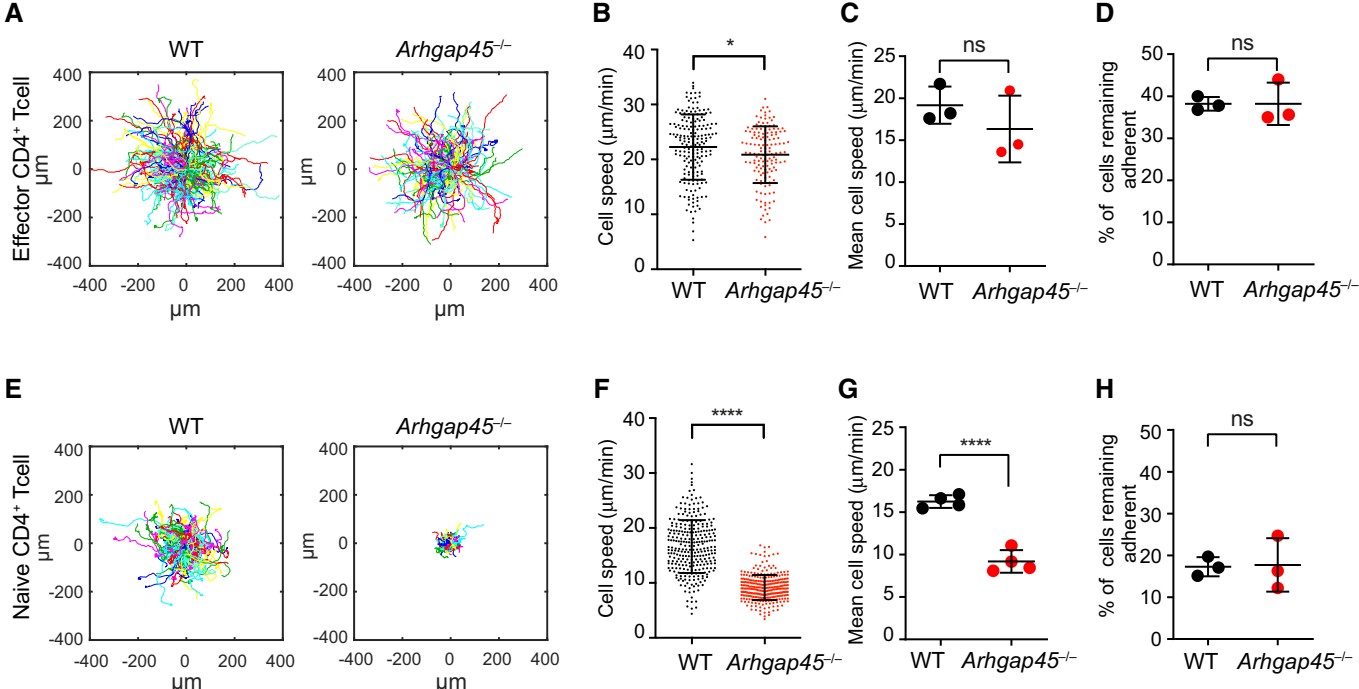

**Figure 5. ARGHAP45 deficiency has no effect on ICAM-1-mediated adhesion and motility of activated T cell but affects speed of naive T cells.**

A   Analysis of migration patterns of WT and *Arghap45*$^{-/-}$ activated T cells on 2D surface coated with ICAM-1. Each track represents the migratory path of individual WT and *Arghap45*$^{-/-}$ activated T cells recorded over 466 s in a single field of view, Trajectories were plotted to a common starting point, and > 200 T cells were recorded per plot using time-lapse microscopy at 10 x magnification.

B   Individual cell speed in an experiment involving 188 WT (black dots) and 144 *Arghap45*$^{-/-}$ (red dots) activated T cells, respectively. Each dot corresponds to the mean speed of a given cell. Bars represent the mean and SD. Three replicates are shown in panel (C).

C   Average cell speed of WT (black dots) and *Arghap45*$^{-/-}$ (red dots) activated T cells. Three independent experiments were performed involving more than 200 cells and each dot correspond to the mean of the speed in one given experiment. Average cell speed was evaluated by dividing the cumulative displacement of cells tracked over 10 seconds by the total time of displacement.

D   Percentage of WT (black dots) and *Arghap45*$^{-/-}$ (red dots) activated T cells remaining adherent after application of a shear stress of 1 dyne/cm$^2$ for 120 s. Three independent experiments were performed involving more than 200 cells and each dot correspond to the mean percentage of remaining adherent cells in one given experiment.

E   Analysis of migration patterns of WT and *Arghap45*$^{-/-}$ naive T cells on 2D surface coated with ICAM-1 and CCL21. Same conditions as in (A).

F   Individual cell speed in an experiment involving 283 WT (black dots) and 305 *Arghap45*$^{-/-}$ (red dots) naïve T cells, respectively. Same conditions as in (B). Four replicates shown in panel (G).

G   Average cell speed of WT cell (black) and *Arghap45*$^{-/-}$ naive T cells (red). Same conditions as in (C).

H   Percentage of WT (black dots) and *Arghap45*$^{-/-}$ naive T cells (red dots) remaining adherent cells after application of a shear stress of 1 dyne/cm$^2$ for 120 s. Same conditions as in (D).

Data information: In (B–D and F–G) mean and SD are shown. *$P \leq 0.033$ (*), **$P \leq 0.002$, ***$P \leq 0.001$, ****$P \leq 0.0001$, unpaired *T*-test.

*Arghap45*$^{-/-}$ mice. When presented *in vitro* by irradiated H-2 K$^b$-positive APC, the N4 peptide induced the proliferation of WT and *Arghap45*$^{-/-}$ OT-I T cells to the same extent (Fig 7A). Therefore, *Arghap45*$^{-/-}$ OT-I T cells do not show measurable defects in their capacity to respond to antigen-laden APC *in vitro*. *Arghap45*$^{-/-}$ OT-I T cells were next evaluated for their ability to respond *in vivo* to physiological numbers of OVA-laden APCs. A portable laser was used to form an array of micropores in the ear epidermis of WT B6 mice whose depth allowed topically applied OVA-loaded XCL1-based vaccine molecules ("vaccibodies") to reach the dermis and specifically target XCR1$^+$ dermal dendritic cells (DC). Under this condition, OVA-laden XCR1$^+$ dermal DC migrate to ear-draining auricular LNs and potently activate adoptively transferred OT-I T cells (Terhorst *et al*, 2015). One day after adoptive transfer of CTV-labeled WT or *Arghap45*$^{-/-}$ OT-I T cells, the ears of WT mice were

subjected to laser treatment and topical application of XCL1-OVA vaccibodies. Four days after immunization, single-cell suspensions were prepared from the auricular LNs, and the extent of OT-I T-cell proliferation determined by CTV dilution (Fig 7B and C). The magnitude of proliferation observed for *Arghap45*$^{-/-}$ OT-I T cells was comparable to that of WT OT-I cells. Therefore, the *Arghap45*$^{-/-}$ OT-I T cells that succeeded homing to LNs showed antigen-driven expansion comparable to their WT OT-I counterparts during DC-triggered *in vivo* immune responses.

Following antigen priming in LNs, effector CD8$^+$ T cells enter the blood to reach infected non-lymphoid tissues. In our model, the mechanical inflammation triggered in the ear dermis by laser treatment vaned rapidly (Terhorst *et al*, 2015) and precluded determining whether *Arghap45*$^{-/-}$ OT-I effector T cells were capable of migrating to the site of vaccination. However, 4 days after

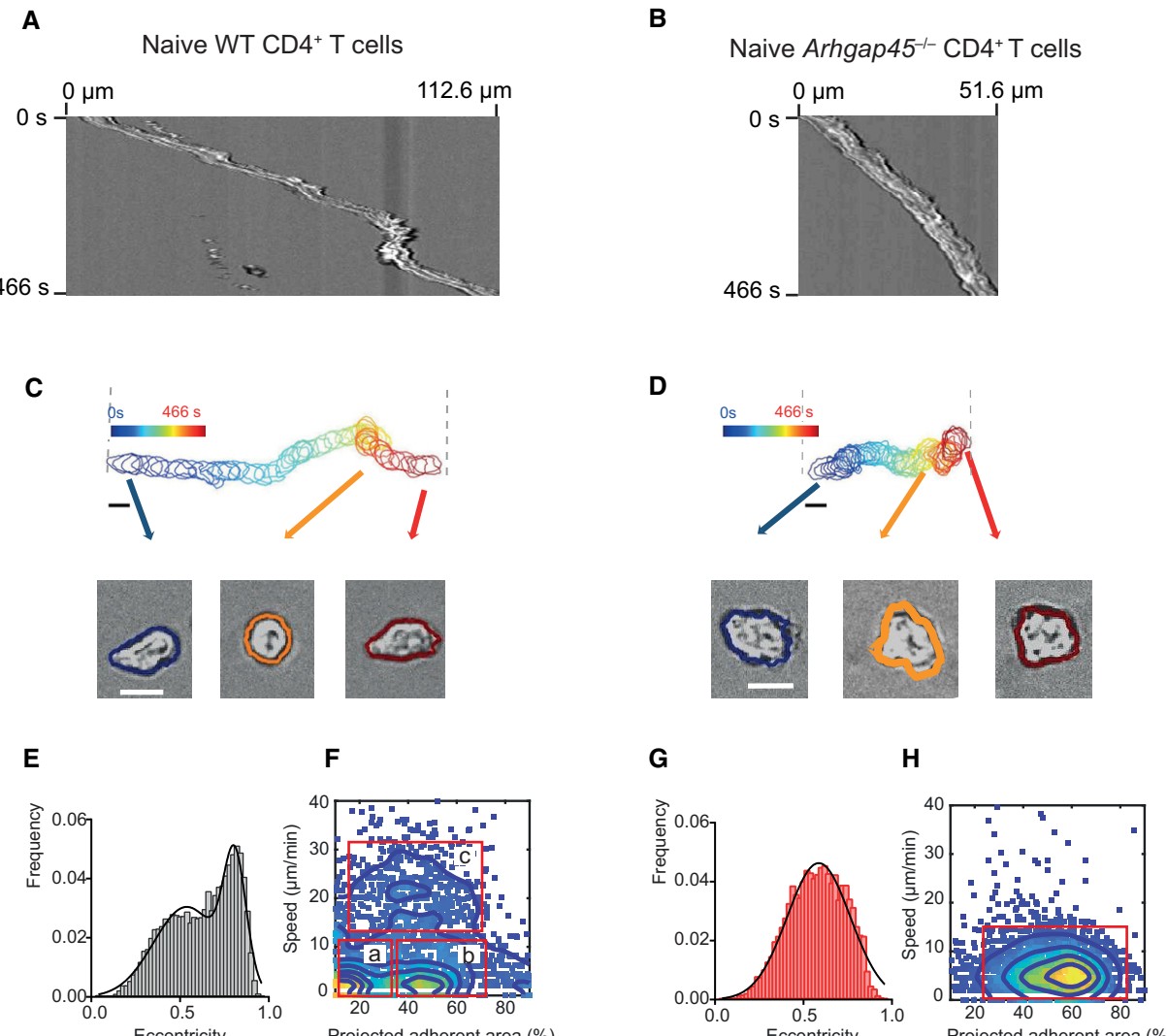

**Figure 6. ARHGAP45 deficiency stabilizes the polarization and decreases the speed of naïve T cells.**

A, B   Representative kymographs of WT (A) and *Arghap45*$^{-/-}$ (B) naive CD4$^+$ T cells on 2D surface coated with ICAM-1 and CCL21.

C, D   Top, representative sequences of cell contours during migration for 466 s of WT (C) and *Arghap45*$^{-/-}$ (D) naive T cells on 2D surface coated with ICAM-1 and CCL21. The color scale highlights each time-resolved shape. Rectangle length, 112.6 μm (WT) and 51.6 μm (*Arghap45*$^{-/-}$). Bar, 10 μm. Bottom, three representative time-resolved shapes and their corresponding calculated contours are shown. Bar, 10 μm.

E, F   Histograms of instant eccentricities (left) and 2D plots of adherent surface versus speed (right) of WT naive T cells on 2D surface coated with ICAM-1 and CCL21. In the 2D plot, the 3 populations discussed in the results have been highlighted: a, hardly motile (instant speed < 5 μm/min) cell population with low (projected adherent area < 20%) adhesion, b, hardly motile (instant speed < 5 μm/min) cell population with high (projected adherent area > 40%) adhesion, and c, highly adherent (projected adherent area > 20%) cell population with elevated speed (ranging between 15 and 30 μm/min).

G, H   Histograms of instant eccentricities (left) and 2D plots of adherent surface versus speed (right) of *Arghap45*$^{-/-}$ naive T cells on 2D surface coated with ICAM-1 and CCL21. In the 2D plot, the single adherent (projected adherent area > 25%) and slow (instant speed < 15 μm/min) cell population is highlighted.

Data information: In (E) and (F), histograms of instant eccentricities correspond to > 60 cells and > 2,000 contours, and fitted by a double Gaussian for WT naive T cells (means = 0.54, 0.8 and SD = 0.18, 0.06) and a single Gaussian for *Arghap45*$^{-/-}$ naïve T cells (Mean = 0.58, SD = 0.18). 2D plots correspond to > 60 cells and > 2,000 events. In (F) and (H), events which combined null motility and adherence over the recorded time have been excluded.

cutaneous immunization, WT and *Arghap45*$^{-/-}$ OT-I T cells with a CTV profile corresponding to that of the effector OT-I T cells that went through the highest rate of division in the auricular LNs were found in the spleen and in non-auricular LNs in numbers commensurable to those of their progeny found in the auricular LNs

(Fig 7B). Therefore, the lack of ARHGAP45 does not measurably affect LN egress of activated CD8$^+$ T cells via efferent lymphatics and their subsequent capacity to re-enter LNs, a finding consistent with the normal chemotaxis of *Arghap45*$^{-/-}$ activated T cells in Transwell migration assay (Fig 4F).

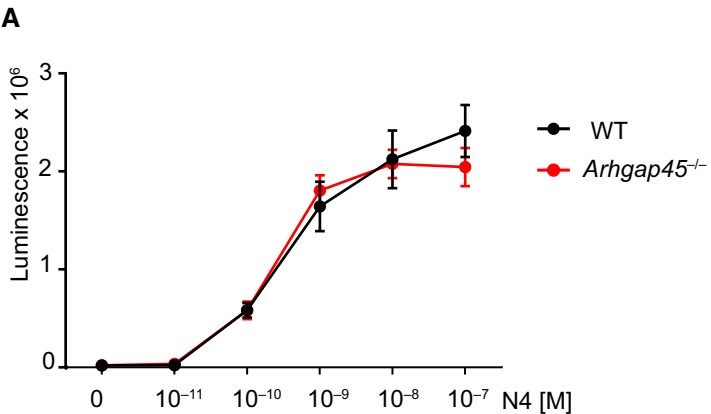

**A**

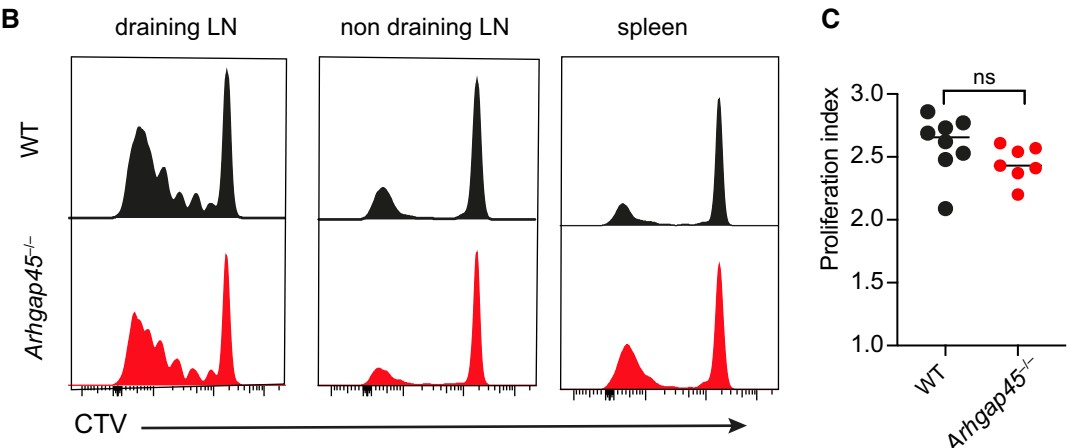

**Figure 7. Normal TCR-dependent activation of *Arhgap45*<sup>−/−</sup> naïve T cells.**

A Irradiated H-2 K<sup>b</sup>- positive spleen cells isolated from T-cell-deficient mice were pulsed for 2 h with the N4 agonist OVA peptide and cultured with CD8<sup>+</sup> T cells purified from WT OT-I (WT) or *Arhgap45*<sup>−/−</sup> OT-I (*Arhgap45*<sup>−/−</sup>) mice. Cell proliferation was measured by luminescence after 48 h. Data are representative of 2 independent experiments with 3 mice per genotype (mean and SEM of triplicate are shown).

B WT mice that received CTV-labeled WT or *Arhgap45*<sup>−/−</sup> OT-I T cells were immunized using laser-assisted, dermal delivery of vaccibodies that target OVA to XCR1<sup>+</sup> DC. Four days after antigen delivery, the extent of OT-I T-cell proliferation was determined by CTV dilution in ear-draining, auricular LNs, in LNs that do not drain the ear, and in the spleen. Data are representative of two experiments with 4 mice per group.

C On the basis of the data shown in (B), the index of OT-I proliferation was calculated using FlowJo software. Each dot represents an individual mouse. Mean and SD are shown. ns, non-significant.

## Competitive repopulation reveals a role for ARHGAP45 in BM engraftment of hematopoietic stem and progenitor cells

Congenic mice co-expressing CD45.1 and CD45.2 were lethally irradiated and reconstituted with a 1 to 1 mixture of BM cells isolated from CD45.1<sup>+</sup> WT mice and from either CD45.2<sup>+</sup> *Arhgap45*<sup>−/−</sup> mice or CD45.2<sup>+</sup> *Arhgap45*<sup>ΔT/ΔT</sup> mice. The resulting CD45.1 WT + CD45.2 *Arhgap45*<sup>−/−</sup> → (CD45.1-CD45.2) and CD45.1 WT + CD45.2 *Arhgap45*<sup>ΔT/ΔT</sup> → (CD45.1-CD45.2) competitive BM chimeras (denoted as *Arhgap45*<sup>−/−</sup> and *Arhgap45*<sup>ΔT/ΔT</sup> BM chimeras, respectively), were analyzed 8 weeks after reconstitution. Control CD45.1 WT + CD45.2 WT → (CD45.1-CD45.2) competitive BM chimeras (denoted as control BM chimeras) were also analyzed in parallel. For each chimera, the percentages of WT CD45.2 cells among WT CD45.2 + WT CD45.1 cells (control BM chimeras), of *Arhgap45*<sup>−/−</sup> CD45.2 cells among *Arhgap45*<sup>−/−</sup> CD45.2 + WT CD45.1 cells (*Arhgap45*<sup>−/−</sup> BM chimeras), and of *Arhgap45*<sup>ΔT/ΔT</sup> CD45.2 cells among *Arhgap45*<sup>ΔT/ΔT</sup> CD45.2 + WT CD45.1 cells (*Arhgap45*<sup>ΔT/ΔT</sup> BM chimeras) were determined (Fig 8). After intravenous infusion, hematopoietic stem and progenitor cells (HSPC) home to the host BM. Using chemokines and integrins, they transmigrate through the sinusoidal walls into the BM extravascular spaces, where they engraft and reconstitute hematopoiesis (Papayannopoulou, 2003). Analysis of *Arhgap45*<sup>−/−</sup> BM chimeras showed that ARHGAP45-deficient HSPC performed less efficiently than WT HSPC as documented by the presence in *Arhgap45*<sup>−/−</sup> BM chimeras of reduced numbers of neutrophils, monocytes, and CD43<sup>+</sup> and CD43<sup>−</sup> early B cells of *Arhgap45*<sup>−/−</sup> origin (Fig 8A). As a result,

the numbers of neutrophils, monocytes, and B cells of $Arhgap45^{-/-}$ origin found in the blood of $Arhgap45^{-/-}$ BM chimera showed a reduction commensurable to that of their BM precursors (Fig 8B). As expected, the development of neutrophils, monocytes, and B cells proceeded unabated in $Arhgap45^{\Delta T/\Delta T}$ BM chimeras (Fig 8C and D). Therefore, under competitive conditions, ARHGAP45 plays a role in BM engraftment of infused HSPC, a result consistent with the expression of ARHGAP45 in long- and short-term repopulating HSPC of the BM (www.immgen.org).

**Competitive repopulation reveals a role for ARHGAP45 in early T-cell development**

T-cell development is dependent on the continuous migration of early T-cell progenitors (ETP) from the BM to the thymus and the CCR7, CCR9, and CXCR4 chemokines play redundant roles in ETP thymus seeding (Scimone *et al*, 2006; Zlotoff *et al*, 2010; Calderon & Boehm, 2011). In $Arhgap45^{-/-}$ BM chimeras, cells of $Arhgap45^{-/-}$ origin were dramatically underrepresented at all the stages of intrathymic T-cell development (Fig 8E). As a consequence, $Arhgap45^{-/-}$-derived CD4$^+$ and CD8$^+$ T cells were strongly reduced in the blood, LNs and spleen of $Arhgap45^{-/-}$ BM chimeras (Fig 8B and F). In marked contrast, in $Arhgap45^{\Delta T/\Delta T}$ BM chimeras, WT and $Arhgap45^{\Delta T/\Delta T}$-derived cells were equally represented at all stages of thymic T-cell development (Fig 8G), whereas $Arhgap45^{\Delta T/\Delta T}$-derived cells T cells were found in reduced numbers in the blood and LNs of $Arhgap45^{\Delta T/\Delta T}$ BM chimeras (Fig 8D and H). The dramatic phenotype observed in the thymus of $Arhgap45^{-/-}$ BM chimeras as compared to that of $Arhgap45^{\Delta T/\Delta T}$ BM chimeras likely reflects the requirement of ARHGAP45 for both ETP generation and subsequent thymus seeding. Along that line, diminished DN1 cell numbers were also noted in $Arhgap45^{-/-}$ mice (Fig 2A) and reduced numbers of common lymphoid progenitors (CLP)—the precursors of ETP—were present in $Arhgap45^{-/-}$ BM as compared to WT BM (Fig EV5A–C). Therefore, the sensitized condition resulting from the competition of ARHGAP45-deficient cells with WT cells demonstrated that ARHGAP45 plays a key role in early T-cell development.

# Discussion

We demonstrated that, under physiological condition, ARHGAP45 plays a critical role in the entry of naïve T and B cells into LNs. ARHGAP45 appeared, however, dispensable for both T- and B-cell development and for the activation of naïve T cells by the antigen-laden DC that reach LNs. It has been suggested that T-cell migration inside LNs occurs in a continuous sliding manner independently of prolonged cycles of adhesion and de-adhesion (Hons *et al*, 2018). Importantly, such mode of low-adhesive intranodal migration differs from the mode of migration occurring during TEM, in which chemokines are thought to activate integrin adhesion in a process known as inside-out signaling. These different migratory modes might thus explain the differential requirement of ARHGAP45 in TEM and intranodal migration of T cells.

After reconstituting lethally irradiated WT mice with a 1 to 1 mixture of BM cells isolated from WT and $Arhgap45^{-/-}$ mice, the competitive repopulation that ensues revealed that ARHGAP45 played additional roles in HSPC BM engraftment and in thymus

seeding. It suggests that an ARHGAP-45-related RHO GAP can compensate for the lack of ARHGAP45 during the almost normal T- and B-cell development that occurred in the non-competitive environment provided in $Arhgap45^{-/-}$ mice. Along that line, our analysis of the ARHGAP45 interactome in T cells showed that ARHGAP45 associates with the RHOA GAP GMIP. GMIP is the closest relative of ARHGAP45 among BAR domain-containing GAP (Amin *et al*, 2016; Carman & Dominguez, 2018), and its pattern of expression overlaps with that of ARHGAP45 (www.immgen.org, https://genevisible.com/search), suggesting the intriguing possibility that GMIP compensates fully (T- and B-cell development) or in part (T- and B-cell entry into LNs) for ARHGAP45 loss in $Arhgap45^{-/-}$ mice.

DP thymocytes have a mean life-span of 3.5 days (Egerton *et al*, 1990) and in $Arhgap45^{\Delta T/\Delta T}$ mice the onset of $Arhgap45$ gene deletion occurred at the DP stage. Therefore, enough ARHGAP45 protein likely persisted in $Arhgap45^{\Delta T/\Delta T}$ DP cells to properly drive their transition to the SP stage. Interestingly, despite the presence of normal numbers of CD4$^+$ and CD8$^+$ SP cells in $Arhgap45^{\Delta T/\Delta T}$ thymi, the blood and LNs of $Arhgap45^{\Delta T/\Delta T}$ mice still showed the same reduction in naïve T-cell numbers as in $Arhgap45^{-/-}$ mice. It suggests that the T-cell lymphopenia observed in $Arhgap45^{-/-}$ mice does not result from the presence of slightly diminished SP numbers but is rather due to the reduced ability of $Arhgap45^{-/-}$ T cells to enter into LNs and receive enough survival-promoting signals on fibroblastic reticular cells (Link *et al*, 2007; Chang & Turley, 2015).

The spherical shape of naïve T lymphocytes circulating in suspension in the blood is due to the contractility of the cortical actomyosin cytoskeleton that lies under the plasma membrane (Stein & Ruef, 2019). Following rolling and sticking on LNs HEV, naïve T cells need to undergo a transition from a spherical to a polarized shape prior to crawl over HEV. Therefore, the poor deformability manifested by $Arhgap45^{-/-}$ naïve T cells might affect this prelude to TEM. Prior to entering LNs, naïve T cells crawl along the HEV luminal surface in search of permissive extravasation sites. Force generation and deformability are both critical to allow squeezing through small openings in the endothelial barrier (Nourshargh *et al*, 2010). Disentangling whether the lack of ARHGAP45 affects the rolling to sticking transition, the crawling along HEV, and/or the extravasation steps remains thus to be established.

In conclusion, our study demonstrates that ARHGAP45 regulates naive T- and B-cell entry into LN. Due to its structural motifs, ARHGAP45 likely delivers its RHO-specific GTPase activity to specific plasma membrane domains that are characterized by the coincident presence of a given curvature and of the CCR7-triggered second messenger DAG. The analysis of the ARHGAP45 interactome (this study) and of its GAP activity in a cell-free system (de Kreuk *et al*, 2013) suggests that ARHGAP45 inhibits RHOA activity. Therefore, ARHGAP45 might contribute to soften the cortical cytoskeleton of naïve T cells thereby increasing their deformability and migratory speed and thus promoting several steps of TEM. Several observations are consistent with such hypothesis. First, the behavior of $Arhgap45^{-/-}$ naïve T cells is reminiscent of that of naïve T cells lacking MYO9B, an F-actin-binding cytoskeletal motor protein with RHO GAP activity (Moalli *et al*, 2018). MYO9B facilitates cell deformability and its absence results in reduced *in vitro* migration toward homeostatic chemokines and lower LN homing *in vivo*. Akin to $Arhgap45^{-/-}$ naïve CD8$^+$ T cells, MYO9B-deficient naïve CD8$^+$ T cells showed normal clonal expansion and effector differentiation in

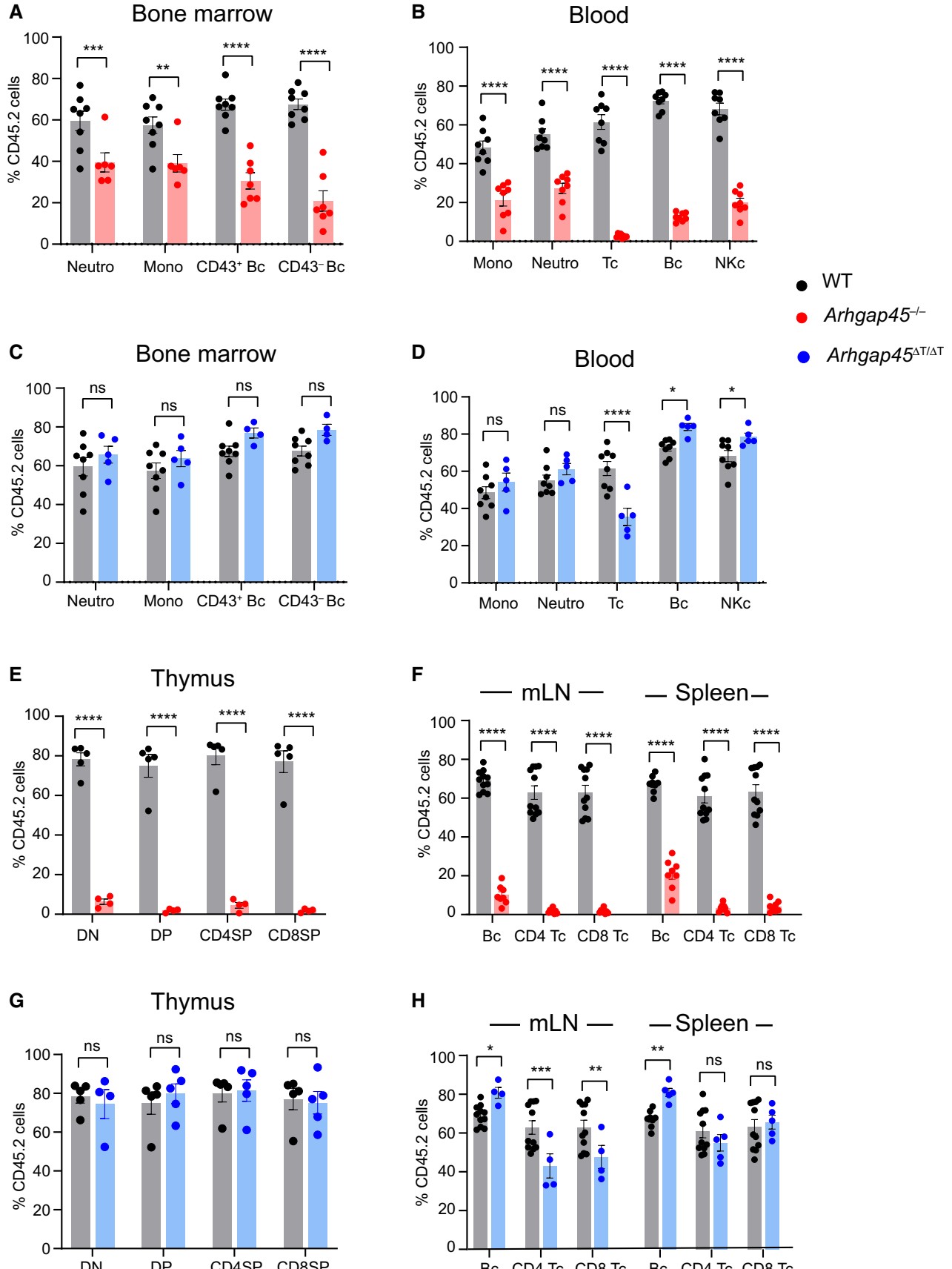

**Figure 8.**

◀

**Figure 8.** *Arhgap45*$^{-/-}$, *Arhgap45*$^{\Delta T/\Delta T}$, and WT progenitors differentially contribute to HSPC BM engraftment and early T-cell development in competitive BM chimeras.

*Arhgap45*$^{-/-}$ and *Arhgap45*$^{\Delta T/\Delta T}$ competitive BM chimeras were analyzed and compared to control (WT) competitive BM chimeras.

A The percentages of WT or *Arhgap45*$^{-/-}$ CD45.2$^{+}$ cells (Key: right side) among CD45.2 + WT CD45.1 cells found in the BM of control and *Arhgap45*$^{-/-}$ competitive BM chimeras are shown.

B The percentages of WT or *Arhgap45*$^{-/-}$ CD45.2$^{+}$ cells among CD45.2 + WT CD45.1 cells found in the blood of control and *Arhgap45*$^{-/-}$ competitive BM are shown.

C The percentages of WT or *Arhgap45*$^{\Delta T/\Delta T}$ CD45.2$^{+}$ cells among CD45.2 + WT CD45.1 cells found in the BM of control and *Arhgap45*$^{\Delta T/\Delta T}$ competitive BM are shown.

D The percentages of WT or *Arhgap45*$^{\Delta T/\Delta T}$ CD45.2$^{+}$ cells among CD45.2 + WT CD45.1 cells found in the blood of control and *Arhgap45*$^{\Delta T/\Delta T}$ competitive BM are shown.

E The percentages of WT or *Arhgap45*$^{-/-}$ CD45.2 + cells among CD45.2 + WT CD45.1 cells found in the thymus of control and *Arhgap45*$^{-/-}$ competitive BM chimeras are shown.

F The percentages of WT or *Arhgap45*$^{-/-}$ CD45.2$^{+}$ cells among the CD45.2 + WT CD45.1 cells found in mesenteric LNs (mLN) and spleen of control and *Arhgap45*$^{-/-}$ competitive BM chimeras are shown.

G The percentages of WT or *Arhgap45*$^{\Delta T/\Delta T}$ CD45.2$^{+}$ cells among CD45.2 + WT CD45.1 cells found in thymus of control and *Arhgap45*$^{\Delta T/\Delta T}$ competitive BM are shown.

H The percentages of WT or *Arhgap45*$^{\Delta T/\Delta T}$ CD45.2$^{+}$ cells among CD45.2 + WT CD45.1 cells found in mesenteric LNs (mLN) and spleen of control and *Arhgap45*$^{\Delta T/\Delta T}$ competitive BM are shown.

Data information: Each dot represents an individual mouse and results in (A)–(H) are representative of two experiments with a total of 5–10 mice per group. Consistent with previous reports (Jafri *et al*, 2017), competitively transplanted mice exhibit a bias in favor of CD45.2 reconstitution. *$P \le 0.033$, **$P \le 0.002$, ***$P \le 0.001$, ****$P \le 0.0001$; unpaired Student's *t*-test. Mean and SD are also shown.

LNs. Second, the ARHGAP45 interactome contains MYO1G, a plasma membrane-associated class I myosin that controls cortical actomyosin cytoskeleton tension and T-cell meandering motility (Olety *et al*, 2010; Gerard *et al*, 2014). Third, GMIP which associates with ARHGAP45 has been shown to inhibit cortical actin polymerization via RHOA inhibition (Aresta *et al*, 2002; Johnson *et al*, 2012; Andrieu *et al*, 2014). Fourth, loss of ARHGAP45 in human umbilical vein endothelial cells increased wound healing and cell migration rates via altered actomyosin contractility (Amado-Azevedo *et al*, 2018). Finally, naïve T cells had a mechanically stiffer cortical cytoskeleton than effector T cells (Thauland *et al*, 2017), and such difference in stiffness might account for our observation that TEM of activated T cell was less dependent on ARHGAP45 than that of naïve T cells.

# Materials and Methods

## Mice

Mice were on C57BL/6 background, sex matched and 6–12 weeks old. They were maintained in specific pathogen-free conditions and used in accordance with institutional committee and European (Marseille) and Chinese (Xinxiang City) guidelines for animal care. OT-I mice have been described (Hogquist *et al*, 1994).

## Generation and validation of mutant mouse constitutively deprived of ARHGAP45

Using CRISPR/Cas9 we established $F_0$ mice with biallelic inactivation of the *Arghap45* gene (Voisinne *et al*, 2019) and after germ-line transmission of an *Arhgap45* allele with a deletion encompassing exon 4 we established mice homozygous for this mutation (Fig EV1A). These mice are denoted as *Arghap45*$^{-/-}$ mice and also known as B6-*Arhgap45*$^{\text{tm1Ciphe}}$.

## Generation of mice with a *lox*P-flanked *Arhgap*45 allele and of mice conditionally deprived of ARHGAP45 in T cells

*Arhgap45*$^{\text{tm1a(EUCOMM)Wtsi}}$ mice were obtained from the Knockout Mouse Project (KOMP) repository (https://www.komp.org/geneinf o.php?geneid = 63136; Fig EV1C). Mice homozygous for the *Arhgap45*$^{\text{tm1a}}$ null allele showed a phenotype identical to that of *Arhgap45*$^{-/-}$ mice with normal thymic cellularity and reduced numbers of naïve T and B cells in LNs (not depicted). The "lacZ-neo" stopper cassette present in the *Arhgap45*$^{\text{tm1a}}$ allele was removed through cross with transgenic mice expressing a FLP recombinase under the control of the actin promoter (Rodriguez *et al*, 2000). It yielded a floxed *Arhgap45* allele denoted as *Arhgap*$^{\text{fl}}$ (also known as *Arhgap45*$^{\text{tm1cCiphe}}$) and in which exons 3 to 10 are bracketed by *lox*P sites (Fig EV1C). Mice homozygous for the *Arhgap*$^{\text{fl}}$ allele have a normal phenotype (not depicted). To achieve specific deletion of ARHGAP45 in T cells, we introduced a CD4-*Cre* transgene (Lee *et al*, 2001) into mice homozygous for the *Arhgap45*$^{\text{fl}}$ allele (Fig EV1C), resulting in mice that are denoted as *Arhgap45*$^{\Delta T}$ and specifically lacked ARHGAP45 expression in T cells (Fig EV1B).

## Cell lines

The Jurkat human leukemic cell line was provided by A. Weiss (University of California San Francisco, CA) and originated from American Type Culture Collection.

## CRISPR-Cas9-based genome editing of Jurkat T cells

The decision to introduce a Twin-Strep-tag coding sequences (OST) at the N-terminus of the ARHGAP45 protein was based on the presence of a putative PDZ-binding domain at its C-terminus (de Kreuk *et al*, 2013). ARHGAP45-deficient Jurkat T cells were first established using the following pairs of sgRNA-specifying oligonucleotide sequences to delete both critical exons 3 and 12 (transcript ID: ENST00000313093.6; exon 3 deletion: 5′-GGGCTCATTTCGAG ACCGCG*GGG*-3′ and 5′-CGAGGGCTCTCTGCCTAGAA*GGG*-3′; exon 12 deletion: 5′-AGTAGGAGATCGTGGCCTGC*GGG*-3′ and 5′-ATA GCTTCCTCCGCCTGCGG *TGG*-3′. A resulting clone (clone 12) that has deleted both exons 3 and 12 and lacked ARHGAP45 protein expression as assessed by Western Blot using an ARHGAP45-specific antibody (Sigma HPA019816) was selected for further study (Fig 1A). A full-length human ARHGAP45 cDNA containing an OST sequence inserted at its 5' end and placed under the control of the CAG promoter was then introduced by homologous recombination into the human *ROSA26* locus (Irion *et al*, 2007; Huang *et al*, 2019)

of clone 12. The resulting Jurkat-ARGHAP45[OST] cells showed levels of ARHGAP45 comparable to WT Jurkat cells and permitted to efficiently affinity purify ARHGAP45[OST] molecules using Sepharose beads coupled to Strep-Tactin (Fig 1A).

## Quantitative interactomics of ARHGAP45[OST] molecules

WT or ARHGAP45[OST] Jurkat cells ($40 \times 10^6$) were left unstimulated or stimulated for 2 min with anti-CD3 antibody (UTHC1) at 37°C. Stimulation was stopped by addition of twice concentrated lysis buffer (100 mM Tris, pH 7.5, 270 mM NaCl, 1 mM EDTA, 20% glycerol and 0.4 % n-dodecyl-β-maltoside) supplemented with protease and phosphatase inhibitors. After 10 min of incubation on ice, cell lysates were centrifuged at 20,000 g for 5 min at 4°C. Post-nuclear lysates were incubated with Strep-Tactin beads for 1.5 h at 4°C. Beads were washed five times with lysis buffer and the bound protein complexes were subsequently eluted by addition of D-Biotin (2.5 mM). Protein extracts were loaded on NuPAGE 4–12% bis-Tris acrylamide gels (Life Technologies) to stack proteins in a single band that was stained with Imperial Blue (Thermo Fisher Scientific) and cut from the gel. Gels pieces were submitted to an in-gel trypsin digestion. Peptides were extracted from the gel and dried under vacuum. Samples were reconstituted with 0.1% trifluoroacetic acid in 4% acetonitrile and analyzed by liquid chromatography (LC)-tandem MS (MS/MS) using a Q Exactive Plus Hybrid Quadrupole-Orbitrap online with a nanoLC Ultimate 3000 chromatography system (Thermo Fisher Scientific™, San Jose, CA). For each biological sample, 5 microliters corresponding to 33% of digested sample were injected in duplicate on the system. After pre-concentration and washing of the sample on a Acclaim PepMap 100 column, peptides were separated on a LC EASY-Spray column at a flow rate of 300 nl/min with a two steps linear gradient (2–22% acetonitrile/H20, 0.1% formic acid for 100 min and 22–32% acetonitrile/H20, 0.1% formic acid for 20 min). For peptide ionization in the EASY-Spray source, spray voltage was set at 1.9 kV and the capillary temperature at 250°C. All samples were measured in a data-dependent acquisition mode. Each run was preceded by a blank MS run in order to monitor system background. The peptide masses were measured in a survey full scan (scan range 375–1,500 m/z, with 70 K FWHM resolution at m/z = 400, and target AGC value of $3.00 \times 10^6$ and maximum injection time of 100 ms). Following high-resolution full scan in the Orbitrap, the 10 most intense data-dependent precursor ions were successively fragmented in HCD cell and measured in Orbitrap (normalized collision energy of 25%, activation time of 10 ms, target AGC value of $1.00 \times 10^5$, intensity threshold $1.00 \times 10^4$ maximum injection time 100 ms, isolation window 2 m/z, 17.5 K FWHM resolution, and scan range 200–2,000 m/z). Dynamic exclusion was implemented with a repeat count of 1 and exclusion duration of 20 s.

## MS data analysis

Raw MS files were processed with MaxQuant software (v.1.6.3.4) for database search with the Andromeda search engine and quantitative analysis. Data were searched against the Homo sapiens entries of the UniProtKB protein database (release Swiss-Prot reviewed 2019_01, 20,412 entries), and the set of common contaminants provided by MaxQuant. Carbamidomethylation of cysteines

was set as a fixed modification, whereas oxidation of methionine and protein N-terminal acetylation were set as variable modifications. Specificity of trypsin digestion was set for cleavage after K or R residues, and two missed trypsin cleavage sites were allowed. The precursor mass tolerance was set to 20 ppm for the first search and 4.5 ppm for the main Andromeda database search. The mass tolerance in tandem MS mode was set to 0.5 Da. Minimum peptide length was set to seven amino acids, and minimum number of unique or razor peptides was set to 1 for validation. The I = L option of MaxQuant was enabled to avoid erroneous assignation of undistinguishable peptides belonging to very homologous proteins. Andromeda results were validated by the target decoy approach using a reverse database, with a FDR value set at 1% at both peptide sequence match and protein level. For label-free relative quantification of the samples, the match between runs option of MaxQuant was enabled with a match time window of 1 min, to allow cross-assignment of MS features detected in the different runs, after alignment of the runs with a time window of 20 min. Protein quantification was based on unique and razor peptides. The minimum ratio count was set to 1 for label-free quantification calculation, and computation of the iBAQ metric was also enabled.

## Determination of cellular protein abundance in the proteome of Jurkat cells

Jurkat cells were washed twice with 1× PBS and cell pellets corresponding to $5 \times 10^6$ cells were lysed in 150 μl of lysis buffer (Tris 50 mM, pH 7.5, EDTA 0.5 mM, NaCl 135 mM, SDS 1%) and kept on ice for 10 min. The samples were frozen and stored at −80°C before sample preparation for MS. 20 μg of protein from each cell lysate were loaded and stacked on NuPAGE 4–12% bis-Tris acrylamide gels (Life Technologies). After staining, proteins bands were cut from the gel and digested with trypsin according to the protocol described above. Protein identification and quantification was performed using liquid chromatography (LC)–tandem mass spectrometry (MS/MS) using an Orbitrap Fusion Lumos Tribrid mass spectrometer in-line with an Ultimate 3000 RSLCnano chromatography system (Thermo Fisher Scientific). For each condition, 3 biological replicates were prepared and each replicate was injected twice on the LC-MS system. In a first step, peptides were concentrated and purified on a Dionex (C18 PepMap100, 2 cm × 100 μm I. D, 100 Å pore size, 5 μm particle size) pre-column using solvent A (0.1% formic acid in 2% acetonitrile). In a second step, peptides were separated on a reverse phase LC EASY-Spray C18 column from Dionex (PepMap RSLC C18, 50 cm × 75 μm I. D, 100 Å pore size, 2 μm particle size) at 300 nl/min flow rate and 40°C. After equilibrating the column using 4% of solvent B (20% water – 80% acetonitrile – 0.1% formic acid), peptides were eluted from the analytical column by a two-step linear gradient (4–22% acetonitrile/H2O; 0.1% formic acid for 130 min and 22–32% acetonitrile/H2O; 0.1% formic acid for 15 min). For peptide ionization in the EASY-Spray nanosource, the spray voltage was set at 2.2 kV and the capillary temperature at 275°C. The Advanced Peak Determination (APD) algorithm was used for real time determination of charges states and monoisotopic peaks in complex MS spectra. The mass spectrometer was used in data-dependent mode to switch consistently between MS and MS/MS. Time between Masters Scans was set to 3 s. MS spectra were acquired in the 400–1,600 m/z range at

a FWHM resolution of 120,000 measured at 200 m/z. AGC target was set at 4.0.105 with a 50 ms maximum injection time. The most abundant precursor ions were selected and collision induced dissociation fragmentation at 35% was performed and analyzed in the ion trap using the "Inject Ions for All Available Parallelizable time" option with a dynamic maximum injection time. Charge state screening was enabled to include precursors with 2 and 7 charge states. Dynamic exclusion was enabled with a repeat count of 1 and a duration of 60 s.

## Data analysis of Jurkat cell proteome

Raw MS files were processed as previously described with MaxQuant software (v.1.6.3.4). Data were searched against the *Homo sapiens* entries of the UniProtKB protein database (release Swiss-Prot reviewed 2020_01, 20,367 entries). Analysis of the proteome of Jurkat cells identified 3,829 protein groups. Protein entries from the MaxQuant "proteinGroups.txt" output were first filtered to eliminate entries from reverse and contaminant databases. Cellular protein abundances were determined from raw intensities using the protein ruler methodology, using the following relationship: protein copies per cell = (protein MS signal × NA × DNA mass)/(M × histone MS signal), where NA is Avogadro's constant, M is the molar mass of the protein and the DNA mass of a diploid human cell is estimated to be 6.5 pg. Cellular protein abundances were log transformed and averaged sequentially over technical and biological replicates.

## Statistical analyses related to MS

From the "proteinGroups.txt" files generated by MaxQuant with the options described above, protein groups with negative identification scores were filtered, as well as proteins identified as contaminants. Protein intensities were log transformed before being normalized across all conditions (condition of stimulation, biological and technical replicates) by the median intensity. Normalized intensities corresponding to different technical replicates were averaged and missing values were replaced after estimating background binding from WT intensities. For each condition of stimulation, we used a two-tailed Welch's t-test to compare normalized protein intensities detected in OST-tagged samples across all biological replicates and WT sample intensities pooled from all conditions of stimulation. Proteins were selected as specifically interacting with ARHGAP45[OST] when both the P-value was below $P = 0.01$ and the corresponding enrichment was greater than two-fold.

## Flow cytometry analysis and antibodies

Stained cells were analyzed using a BD LSRII™ or a FACSymphony™ flow cytometer and a FACSDiva 9.1 software (BD). Cell viability was evaluated using SYTOX™ Blue Dead Cell Stain (Invitrogen) or DAPI (Merck). The following antibodies were used. CD4 (RMA-5); CD8α (53–6.7), CD5 (53–7.3), CD16/CD32 (2.4G2), CD19 (1D3), CD24 (M1/69), CD25 (PC61), CD27 (LG.3A10), CD34 (RAM34), CD43 (S7), CD44 (IM7), CD45.1(A20), CD45.2 (104), CD45R (RA3-6B2), CD62L (Mel-14), BP-1 (6C3), IgM (R6-60.2), IgD (11-26c.2a), Ly-6G (clone 1A8), Ly-6C (AL-21), Sca-1 (Ly-6A/E, D7), TCRβ (H57-597), TCRγδ (GL3), Ter119 (TER-119), all from BD Biosciences.

CD11b (M1/70), CD3 (17A2), CD150 (TC15-12F12.2), CD135 (A2F10), CD161c (PK136), CD11c (N418), CD45 (30-F11), CD19 (6D5), and CD197 (CCR7, 4B12) all from BioLegend, CD127 (A7R34) and CD117 (2B8) both from eBioscience, and LFA-1 (CD11a, 121/7) from Invitrogen.

## Estimation of LN and spleen homing rate

LN and spleen entry rates were determined as previously described (Arbones *et al*, 1994; Arnon *et al*, 2011). Splenocytes from WT or *Arhgap45*[−/−] mice were labeled with 0.2 μM CellTracker™ Red CMTPX dye (Invitrogen) or 0.5 μM CellTrace™ Violet proliferation kit (Molecular Probes). WT and *Arhgap45*[−/−] mice were mixed at 1 to 1 ratio and $40 \times 10^6$ injected i.v. in each recipient mice. After 4 h, cells were recovered from LNs and spleen and the ratio of CMTPX- and CTV-labeled T and B cells measured by flow cytometry.

## Transwell migration assay

Migration was studied with a Transwell tissue culture system (Corning) containing insert membrane with 5 μm pores. $0.5 \times 10^6$ isolated CD4[+] T cells (EasySep™ Mouse CD4[+] T-cell isolation Kit, STEMCELL™) or isolated CD8[+] T cells (Dynabeads® Untouched™ Mouse CD8 cells, Invitrogen) from WT, *Arhgap*[−/−] or *Arhgap45*[ΔT/ΔT] mice were added to the upper chamber and allowed to transmigrate for 2 h at 37°C in the presence or absence of 100 μg/ml of CCL19, CCL21, or CXCL12 (all from PeproTech®) in the lower chamber. When specified, T cells were activated for 2 days with plate-bound anti-CD3 at 3 μg/ml (145-2C11; Exbio) and soluble anti-CD28 at 1 μg/ml (37-51; Exbio), rested for 2 days in IL-2 prior to be tested in Transwell migration assay. Cells that had migrated to the bottom chamber were counted using flow cytometry after adding 10 μl of cell counting beads (CountBright™, Invitrogen). Migration rate was measured as percentage of migrated cells relative to total input ([no. of migrating cells/no. of input cells] × 100). $0.5 \times 10^6$ purified B cells (Dynabeads® Untouched™ Mouse B cells, Invitrogen) were similarly analyzed for their ability to transmigrate for 4h in the presence or absence of 500 μg/ml of the specified chemokines in the lower chamber. When specified, insert membrane was coated with 10 μg/ml of recombinant mouse ICAM-1 (R&D Systems) or PBS 1× for 4 h prior to the migration test.

## Generation of BM competitive chimeras

Six- to 8-week-old B6 CD45.1 x CD45.2 mice were lethally irradiated with two doses of 3 Gy given 4 h apart and then injected i.v. with $4 \times 10^6$ BM cells. BM cells were obtained from femurs and tibias of WT B6 CD45.1 mice or of WT, *Arhgap45*[−/−], or *Arhgap45*[ΔT/ΔT] CD45.2 mice.

## Antigen-induced OT-I T-cell proliferation

CD8[+] T cells were purified from WT and *Arhgap45*[−/−] OT-I mice by immunomagnetic negative selection using Dynabeads® Untouched™ Mouse CD8 cells, (Invitrogen). Purified T cells were stimulated with irradiated H-2 K[b]- positive spleen cells isolated from T-cell-deficient mice and pulsed for 2 h with the N4 agonist OVA peptide or with PMA and ionomycin. After 48 h of culture, T-cell

proliferation was assessed with CellTiter-Glo® Luminescent Cell Viability Assay (Promega). The resulting luminescence, which is proportional to the ATP content of the culture, was measured with a Victor 2 luminometer (Wallac, Perkin Elmer Life Science).

### OT-I T-cell responses to laser-assisted intradermal delivery of vaccines

Vaccibodies containing the XCL1 chemokine as the targeting unit and the OVA as the antigen were delivered into the ear epidermis of wild-type CD45.1 B6 mice using a P. L. E. A. S. E portable laser (Terhorst *et al*, 2015). OT-I T cells were isolated from spleen of WT and *Arhgap45*$^{-/-}$ OTI mice (see above) and their purity determined by staining with CD4, CD8, CD5, and TCR Vα2. Purified OT-I cells were labeled with CellTrace™ Violet proliferation kit (Molecular Probes) by resuspending them in PBS containing 2.5 μM CTV for 20 min at 37°C. $10^6$ CTV-labeled OT-I cells were adoptively transferred into the specified mice. At the indicated times, single-cell suspensions were prepared from the spleen, the auricular LNs draining the immunized ears, and the mesenteric LNs that do not drain the skin. OT-I cells were analyzed by FACS for their CTV content.

### Statistical analyses related to immunophenotyping

Statistical significance was determined using two-tailed Student's *t*-tests with Welch's correction, one- or two-way ANOVA test, with or without Sidak correction for multiple testing, as specified in individual figure legends. Points in graphs indicate either individual mice and lines indicate means or medians. In bar graphs, bars indicate means, and error bars indicate SD or SEM as specified in individuals figure legends. Analysis was performed using GraphPad Prism software. *P* values lower than 0.033 were considered statistically significant.

### Migration assays

Migration assays were performed in ibidi channels (uncoated μ-Slide VI 0.4 channels, ibidi, Martinsried, Germany) precoated overnight at 4°C with either pure recombinant mouse ICAM-1 Fc chimera (R&D Systems, Minneapolis, MN, USA) at 10 μg/ml or a with mix of recombinant ICAM-1 Fc chimera at 10 μg/ml and CCL21 (Miltenyi Biotec Company, CA, USA) at 1 μg/ml. The channels were then incubated for 15 min at room temperature with 4% bovine serum albumin (BSA) (Axday, Dardilly, France) and finally rinsed three times with phosphate-buffered saline (PBS) (Gibco by Thermo Fisher Scientific, Waltham, MA, USA). Control and ARHGAP-KO naïve mouse T cells ($5 \times 10^4$ cells/ml and > 97% viable as determined by Trypan blue exclusion) were plated in channels and placed in a humidified incubator at 37°C and 5% $CO_2$ for 10 min before observation. Cells were then imaged at 37°C with an inverted Zeiss Z1 automated microscope (Carl Zeiss, Oberkochen, Germany) equipped with a Snap HQ CCD camera (Photometrics, Tucson, AZ), pE-300 white LED microscope illuminator (CoolLED, Andover, UK) piloted by μManager (Edelstein *et al*, 2014).

### Cell tracking

Cells were tracked with a home-made program developed with MATLAB software (The MathWorks, Natick, MA, USA) as

previously described (Valignat *et al*, 2013). Briefly, the program works on binarized images of a video microscopy time sequence. To analyze a large number of cells per image (~ 500 cells), bright field images with low magnification objective (Plan-Neofluar 10×/0.3 objective, Carl Zeiss, Oberkochen, Germany) were collected every 10 s for 17 min. Each frame was divided by a background frame and intensity to homogenize all frames of a given stack, and finally binarized so that each individual cell appeared as a homogeneous white spot in a surrounding black background. Binarized images were then used to determine the trajectory and speed of each cell. In each experiment, the mean speed of crawling cells was calculated after exclusion of immobile cells (i.e., cells with curvilinear displacement below 10 μm during 1,000 s). To assess the precise morphology and the adhesion fingerprint of individual cells, a sequence of bright field and reflection interference contrast microscopy (RICM) images were taken every 2s for 14 min with a ×63 objective (Neofluar 63/1.25 antiflex, Carl Zeiss, Oberkochen, Germany). All raw images were also treated by background division and intensity normalization. Then, bright field images were binarized using the Pixel classification workflow of the Ilastik software[79] and further analyzed using the image processing toolbox of Matlab to obtain the circularity and eccentricity of the cell contour. RICM images were inverted and thresholded. The adhesion fingerprint (appearing dark in raw images and white in treated images) was then detected for each cell. The adhesion for each cell was finally quantified by dividing the area of the contact fingerprint (obtained from RICM images) by the projected area of the whole cell (obtained from bright field images).

## Data availability

The mass spectrometry proteomics data have been deposited at the ProteomeXchange Consortium via the PRIDE partner repository (http://www.ebi.ac.uk/pride) with the dataset identifiers PXD018016.

**Expanded View** for this article is available online.

### Acknowledgements

We thank M. Bajenoff and S. Sarrazin for discussion, Rong Huang for preparing Fig 1A, Raana Ramouz-Charpentier for technical help, and the EUCOMM Program for *Arhgap45*$^{\text{tm1a(EUCOMM)Wtsi}}$ mice. This work was supported by CNRS, INSERM, the European Union's Horizon 2020 research and innovation program (grant agreement n° 787300 (BASILIC) to B. M.), the MSDAVENIR Fund (to B. M.), the Investissement d'Avenir program PHENOMIN (French National Infrastructure for mouse Phenogenomics; ANR-10-INBS-07, to B. M.), the National Natural Science Foundation of China (grants n° 81471595 and 32070898 to Y. L. and 81901573 to L. H), the 111 project D20036 (to Y. L.), and by fellowships from Fondation pour la Recherche Médicale (L. G.). the Agence Nationale de la Recherche (RECRUTE: ANR-15-CE15-0022 and ILIAAD: ANR-18-CE09-0029, to O. T), the Excellence Initiative of Aix-Marseille University-A*MIDEX, a French "Investissements d'Avenir" program (to M.-P. V.), The Marseille Proteomics facility is supported by IBISA (Infrastructures Biologie Santé et Agronomie), Plan Cancer, Canceropôle PACA, Région Sud Provence-Alpes-Côte d'Azur, Institut Paoli-Calmettes, Centre de Recherche en Cancérologie de Marseille, Plate-forme Technologie Aix-Marseille and Fonds Européen de Développement Régional.

## Author contributions

BM, MM, and YL conceived the project. LH, M-PV, LG, LZ, and FZ generated and analyzed the data with advices from VLG, SH, HW, and OT. SA, LC, and RR performed the mass spectrometry analysis. EF and BB provided vaccibodies. BM and MM wrote the manuscript with the help of RR and YL, OT, and M-PV.

## Conflict of interest

The authors declare that they have no conflict of interest.

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
