## [Review Process File · EMBO Reports]

ARHGAP45 controls naïve T and B cell entry into lymph nodes and T-cell progenitor thymus-seeding

Le He, Marie-Pierre Valignat, Lichen Zhang, Lena Gelard, Fanghui Zhang, Valentin Le Guen, Stephane Audebert, Luc Camoin, Even Fossum, Bjarne Bogen, Hui Wang, Sandrine Henri, Romain Roncagalli, Olivier Theodoly, Yinming Liang, Marie Malissen and Bernard Malissen
DOI: 10.15252/embr.202052196

Corresponding author(s): Bernard Malissen (bernardm@ciml.univ-mrs.fr), Marie Malissen (malissen@ciml.univ-mrs.fr), Yinming Liang (yinming.liang@foxmail.com)

Review Timeline:

Submission Date:	30th Nov 20
Editorial Decision:	1st Dec 20
Revision Received:	14th Dec 20
Editorial Decision:	7th Jan 21
Revision Received:	20th Jan 21
Accepted:	25th Jan 21

Editor: Achim Breiling

Transaction Report: This manuscript was transferred to EMBO reports following peer review at The EMBO Journal.

Dear Dr. Malissen,

Thank you for transferring your manuscript to EMBO reports. I now went through your manuscript and the referee reports from The EMBO Journal (attached below). Both referees acknowledge that the findings are of interest. Nevertheless, they have raised a number of concerns and suggestions to improve the manuscript, or to strengthen the data and the conclusions drawn.

EMBO reports emphasizes novel functional over detailed mechanistic insight, but asks for strong in vivo relevance of the findings, and clear experimental support of the major conclusions. Thus, we will not require addressing points regarding more mechanistic details experimentally. However, it will be necessary that in a revised manuscript you address all points questioning the main conclusions of the study, and all technical concerns, or points regarding the experimental designs, model systems used, or data presentation.

Given the constructive referee comments, we would like to invite you to revise your manuscript with the understanding that all referee concerns must be addressed in the revised manuscript and/or in a detailed point-by-point response. Acceptance of your manuscript will depend on a positive outcome of a second round of review. It is EMBO reports policy to allow a single round of revision only and acceptance of the manuscript will therefore depend on the completeness of your responses included in the next, final version of the manuscript.

Revised manuscripts should be submitted within three months of a request for revision. We are aware that many laboratories cannot function at full efficiency during the current COVID-19/SARS-CoV-2 pandemic and we have therefore extended our 'scooping protection policy' to cover the period required for full revision. Please contact me to discuss the revision should you need additional time, and also if you see a paper with related content published elsewhere.

- 1) a .docx formatted version of the final manuscript text (including legends for main figures, EV figures and tables), but without the figures included. Please make sure that changes are highlighted to be clearly visible. Figure legends should be compiled at the end of the manuscript text.
- 2) individual production quality figure files as .eps, .tif, .jpg (one file per figure), of main figures and EV figures. Please upload these as separate, individual files upon re-submission.

The Expanded View format, which will be displayed in the main HTML of the paper in a collapsible format, has replaced the Supplementary information. You can submit up to 5 images as Expanded View. Please follow the nomenclature Figure EV1, Figure EV2 etc. The figure legend for these

should be included in the main manuscript document file in a section called Expanded View Figure Legends after the main Figure Legends section. Additional Supplementary material should be supplied as a single pdf file labeled Appendix. The Appendix should have page numbers and needs to include a table of content on the first page (with page numbers) and legends for all content. Please follow the nomenclature Appendix Figure Sx, Appendix Table Sx etc. throughout the text, and also label the figures and tables according to this nomenclature.

For more details please refer to our guide to authors:

See also our guide for figure preparation:

http://wol-prod-cdn.literatumonline.com/pb-assets/embosite/EMBOPress_Figure_Guidelines_061115-1561436025777.pdf

4) a complete author checklist, which you can download from our author guidelines (<https://www.embopress.org/page/journal/14693178/authorguide>). Please insert page numbers in the checklist to indicate where the requested information can be found in the manuscript. The completed author checklist will also be part of the RPF.

Please also follow our guidelines for the use of living organisms, and the respective reporting guidelines: <http://www.embopress.org/page/journal/14693178/authorguide#livingorganisms>

5) that primary datasets produced in this study (e.g. RNA-seq, ChIP-seq and array data) are deposited in an appropriate public database. This is now mandatory (like the COI statement). If no primary datasets have been deposited in any database, please state this in this section (e.g. 'No primary datasets have been generated and deposited').

The accession numbers and database should be listed in a formal "Data Availability " section (placed after Materials & Methods) that follows the model below. Please note that the Data Availability Section is restricted to new primary data that are part of this study.

Data availability

6) We strongly encourage the publication of original source data with the aim of making primary data more accessible and transparent to the reader. The source data will be published in a separate source data file online along with the accepted manuscript and will be linked to the relevant figure. If you would like to use this opportunity, please submit the source data (for example scans of entire gels or blots, data points of graphs in an excel sheet, additional images, etc.) of your key experiments together with the revised manuscript. If you want to provide source data, please include size markers for scans of entire gels, label the scans with figure and panel number, and send one PDF file per figure.

8) Regarding data quantification and statistics, please specify, where applicable, the number "n" for how many independent experiments (biological or technical replicates) were performed, the bars and error bars (e.g. SEM, SD) and the test used to calculate p-values in the respective figure legends. Please provide statistical testing where applicable, and also add a paragraph detailing this to the methods section. See: <http://www.embopress.org/page/journal/14693178/authorguide#statisticalanalysis>

9) Please provide the abstract written in present tense.

10) Please note our new reference format:
<http://www.embopress.org/page/journal/14693178/authorguide#referencesformat>

I look forward to seeing a revised version of your manuscript when it is ready. Please let me know if you have questions or comments regarding the revision.

Kind regards,

Achim

Achim Breiling
Editor
EMBO Reports

Referee #1:

This study shows that ARHGAP45 predominantly associates with RhoA amongst small G proteins

and also interacts with some other partners in (Jurkat) T cells. KO mice are generated and shown to have reduced T and B cells in their LNs and PPs. Mixed BM chimeras also reveal reduced competitiveness in reconstitution of lymphoid lineages. In vitro studies show strongly reduced migration of KO naïve (but not activated) lymphocytes to several chemokines. Imaging of cells migrating on ICAM1 in the presence of chemokine provides evidence that the GAP regulates T cell deformability during 2D movement. Somewhat unexpectedly, the ability of T cells to respond to antigen in LNs is found to be intact. Overall, this study provides substantial new insight regarding the role of ARHGAP45 in lymphocytes, establishing a critical role in naïve lymphocyte entry from blood into LNs and PPs.

Specific comments:

1. The exact role of ARHGAP45 during lymphocyte entry into LN and PP is inferred but not directly addressed. Given that there is some evidence that Rho can be involved in chemokine triggered adhesion of lymphocytes under shear (e.g. Laudanna et al., Science 1996 271, 981), it seems possible that the defect is in the rolling to sticking transition rather than in the TEM step that the authors suggest (or at both steps). If the authors are not in a position to perform intravital video microscopy of lymphocyte rolling and sticking on LN HEV (the best approach to resolve this issue but arguably beyond the scope of this study), they might still be able to provide some information on this point by examining what fraction of the KO vs WT cells in the LN are blood exposed. If adhesion to HEV is still occurring but the cells are not entering, then they might be transiently accumulated on the HEV. This can sometimes be measured by performing intravascular antibody labeling (e.g. 2min in vivo with anti-CD45-PE) of short term transferred cells to determine (by FACS) what fraction of the cells in the LN are in a blood exposed compartment. If the cells are able to adhere normally (as the authors suggest) but are inefficient at TEM, the fraction of transferred cells in the LN cell suspension that are intravascularly labeled will be elevated (e.g. Arnon et al., Science 2011 333, 1898). On the other hand, if the rolling to sticking transition is the main step that is affected, then the adherent blood exposed fraction may not be elevated. Another method to look at this would be to examine the distribution of cells in LN sections using microscopy (e.g. Kanda et al., Nature Immunol 2008 9, 415).
2. The normal T cell response in vivo in the draining LN suggests T cell motility in the LN may be less affected than suggested by the defective migration to chemokines in vitro. The authors might comment about this. If sections showing transferred T cell distribution in WT spleen and LN are available, these could be informative.
3. Minor: it is stated that 'a similar trend (to LN) occurred in Arhgap45 KO PP' but the data show a stronger phenotype.

Referee #2:

This manuscript provides a detailed and thorough analysis comparing T cells and B cells in and derived from wt versus arhgap45^{-/-} mice. In a separate and unlinked proteomic experiment, they describe the proteins associated with ARHGAP45 isolated from Jurkat T cells. While the data for the most part well controlled and reproducible, there is no information on the mechanistic basis for any of the analyses on T cells and B cells from the mice. Are all these phenotypes due to a change in RhoA activation or are they due to another function of ARHGAP45 or other interacting protein? There is no attempt to relate any of the interesting but preliminary proteomic analysis to the descriptive phenotypic analysis. Without any indication of the mechanism this has limited interest to scientists working on either Rho GTPase signaling or T/B cell signaling/function.

Points to consider for revisions:

1. English needs improving (grammar and choice of some words).
2. Abstract: 20 RhoGAPs (need to be specific because of Ras, Rab etc. GAPs)
3. Abstract: 'non-competitive setting/competitive setting' needs rewording - these are not familiar phrases for a non-specialist reader.
4. Introduction: The second paragraph needs references added throughout - there are none. There are 20 Rho GTPases in mammals, which are subdivided into 8 subgroups. GEF stands for guanine nucleotide exchange factor.
5. Proposed RhoA specificity of ARHGAP45 - what about the highly homologous RhoB and RhoC? RhoC levels could be lower than RhoA because ARHGAP45 is a better GAP for RhoC than RhoA, hence RhoC is not bound efficiently because the GTP has been hydrolysed and it has dissociated from ARHGAP45.
6. YWHAQ is 14-3-3 θ : it is important to state this when YWHAQ is first mentioned.
7. The mass spectrometry approach recognizes proteins that interact directly with ARHGAP45 and proteins that are associated with it indirectly via one or more intermediary proteins. It is not possible to distinguish between these two possibilities so 'interact' should be changed to 'associate' throughout the section on the 'interactome' (p. 5/6).
8. For all graphs, absolute p values should be provided, not n.s. or asterisks. This allows readers to judge for themselves whether two values should be considered different or not. If numbers analysed vary between conditions in one experiment, then the numbers should be written above the condition directly, not given as a range in the figure legend. It is then much clearer which experimental conditions had fewer cells/mice.
9. Figure 2: it is preferable to show the results for each mouse as a dot (as in Figure 3) and then the mean and SD on top of this. A good way to present this is to use different colors of dots for mice from each of the experiments, so that the reader can see if there is variation in the absolute numbers between different days or mice.
10. Figure 3B: It is not possible to compare results from 3 different experiments with arhgap45^{-/-} mice with controls from only one experiment. The arhgap45^{-/-} mice analysed on the same day as the control mice should be shown. The other mice from different days could be shown as separate bars for each day to indicate what the values were but not compared statistically.
11. Figure 3D and Figure 4: the results from all the mice analysed should be shown here - it is not clear what 'representative' means. Why not show all the results - if they vary between different experiments this can be illustrated by using dots of different colors for each experiment?
12. Figure 5: As well as showing the mean speed of all cells from each experiment, it is essential to show the range of all cell speeds for each experiment with one dot for each cell, indicating the median cell speed. This is a standard way to represent cell speeds. It could be that mean speed for all cells is not different, but the range of cell speeds varies more under some conditions (as suggested in Figure 6). In the movies a proportion of cells are almost stationary, and this number could vary e.g. it could be higher for arhgap45^{-/-} cells or could vary between individual experiments. Although the authors claim there is no difference between wt and arhgap45^{-/-} activated T cells, the cells in the movies look different in their behavior and polarity. Notably, there are far fewer cells in the movies with arhgap45^{-/-} T cells than control cells (both activated and naïve). Is this because fewer arhgap45^{-/-} cells adhere under static conditions to ICAM1 or because they are less viable? In either case, quantification is required. (The flow assays do not provide the relevant information on adhesion under static conditions).
13. Movie legends should state how many frames in the movie and time between each frame.
14. Transwell migration assays. It is surprising that 5 μ m pore size was used - for T and B cells it is standard to use 3 μ m pore size because of their small size compared to other cell types. The authors should repeat at least some of the experiments with 3 μ m pores.

We are grateful to the two Reviewers that provided feedback for the time and consideration shown to our manuscript. We are also gratified by their generally positive remarks about our paper. Both Reviewers brought up a number of points that we have directly addressed in the revised manuscript. Below, we provide a detailed response to the Reviewers' comments, along with a description of associated changes to the manuscript. All changes in response to the reviewers are highlighted in yellow in the revised manuscript. Our paper has been transferred from EMBO Journal to EMBO Reports. In contrast to EMBO Journal, 'EMBO Reports emphasizes novel functional over detailed mechanistic insight'. Accordingly, and as suggested by the EMBO Reports Editor, we have not addressed points regarding more mechanistic details experimentally.

Referee #1:

This study shows that ARHGAP45 predominantly associates with RhoA amongst small G proteins and also interacts with some other partners in (Jurkat) T cells. KO mice are generated and shown to have reduced T and B cells in their LNs and PPs. Mixed BM chimeras also reveal reduced competitiveness in reconstitution of lymphoid lineages. In vitro studies show strongly reduced migration of KO naïve (but not activated) lymphocytes to several chemokines. Imaging of cells migrating on ICAM1 in the presence of chemokine provides evidence that the GAP regulates T cell deformability during 2D movement. Somewhat unexpectedly, the ability of T cells to respond to antigen in LNs is found to be intact. Overall, this study provides substantial new insight regarding the role of ARHGAP45 in lymphocytes, establishing a critical role in naïve lymphocyte entry from blood into LNs and PPs.

Specific comments

1. The exact role of ARHGAP45 during lymphocyte entry into LN and PP is inferred but not directly addressed. Given that there is some evidence that Rho can be involved in chemokine triggered adhesion of lymphocytes under shear (e.g. Laudanna et al., Science 1996 271, 981),

it seems possible that the defect is in the rolling to sticking transition rather than in the TEM step that the authors suggest (or at both steps). If the authors are not in a position to perform intravital video microscopy of lymphocyte rolling and sticking on LN HEV (the best approach to resolve this issue but arguably beyond the scope of this study), they might still be able to provide some information on this point by examining what fraction of the KO vs WT cells in the LN are blood exposed. If adhesion to HEV is still occurring but the cells are not entering, then they might be transiently accumulated on the HEV. This can sometimes be measured by performing intravascular antibody labeling (e.g. 2min in vivo with anti-CD45-PE) of short term transferred cells to determine (by FACS) what fraction of the cells in the LN are in a blood exposed compartment. If the cells are able to adhere normally (as the authors suggest) but are inefficient at TEM, the fraction of transferred cells in the LN cell suspension that are intravascularly labeled will be elevated (e.g. Arnon et al., Science 2011 333, 1898). On the other hand, if the rolling to sticking transition is the main step that is affected, then the adherent blood exposed fraction may not be elevated. Another method to look at this would be to examine the distribution of cells in LN sections using microscopy (e.g. Kanda et al., Nature Immunol 2008 9, 415).

Unfortunately, we are not in a position to perform intravital video microscopy of lymphocyte rolling and sticking on LN HEV. In Europe, Professor Wolfgang Kastentmüller is a leader in intravital video microscopy of lymphocytes. Therefore, we contacted him 2-years ago to set up a collaboration on Arhgap45^{-/-} mice but at that time he was just in the process of moving his lab to Würzburg. Due to the COVID 19 lock-down over Spring 2020 and the need to diminish by 1/3 our mice colonies, we failed re-initiating this collaboration in 2020. However, we were aware of the very relevant paper quoted by Reviewer #1 (Arnon et al., 2011). More specifically, panels E and F in Figure 2 of the paper by Arnon and colleagues show that it is possible ‘to measure the fraction of B cells that entered the LN versus remained within the HEV’. Accordingly, at the end of the lockdown, we performed on Arhgap45^{-/-} and WT mice a series of experiments identical to those of Arnon and colleagues. As originally reported by Arnon and colleagues (see Figure 2E in Arnon’s paper), we observed that rather small numbers of WT and Arhgap45^{-/-} T cells (from 10 to 50) remained blood-exposed at the end of the experiments. The small numbers of T cells that remained blood-exposed limited the quality of the statistics and led us to keep those data out of the paper. Although Reviewer #1, consider that resolving this issue is ‘arguably beyond the

scope of this study', we still attempted to repeat such experiments with the *Arhgap45*^{-/-} mice that remained available to us, and specifically attempted to sensitize it by injecting purified T cells from CD45.1 WT and CD45.2 *Arhgap45*^{-/-} mice instead of injecting total spleen cells as originally reported by Arnon and colleagues and as we did in our first series of experiments. Accordingly, 10×10^6 T cells from each genotype were mixed and then injected intravenously into CD45.1 X CD45.2 recipient mice. After 30 minutes, 3 μ g of Alexa700-conjugated-anti-CD45 antibody were injected intravenously. Mice were sacrificed after 3 minutes and their peripheral lymph-nodes collected. T cells were then isolated and analyzed by flow cytometry. For the attention of Reviewer #1, we have prepared a figure (see below) which shows the result of this recent experiment that involved four injected CD45.1 X CD45.2 recipient mice. Panel A shows flow cytometry analysis of 30-min transferred WT and *Arhgap45*^{-/-} T cells labeled for 3 min *in vivo* with Alexa700-conjugated-anti-CD45 antibody. Panel B shows the ratio of *Arhgap45*^{-/-} T cell numbers relative to WT T cell numbers that were either inside the LN and protected from staining with Alexa700-conjugated-anti-CD45 antibody (A700⁻) or blood-exposed (A700⁺), and identified as in panel A. The p value between the two conditions is non-significant. Consistent with the presence of 2.2-fold diminished numbers of naïve T cells in *Arhgap45*^{-/-} LN (Figure 2 of our manuscript), 2.5-fold less *Arhgap45*^{-/-} T cells were protected from staining as compared to WT T cells. Likewise, 3.1-fold less *Arhgap45*^{-/-} T cells were blood-exposed as compared to WT T cells. This last result suggests that the rolling to sticking transition is likely the main step affected during TEM of *Arhgap45*^{-/-} T cells.

Accordingly, in the revised Discussion we mitigated our original conclusions as follow:

‘The spherical shape of naïve T lymphocytes circulating in suspension in the blood is due to the contractility of the cortical actomyosin cytoskeleton that lies under the plasma membrane (Stein and Ruef, 2019). Following rolling and sticking on LN HEV, naïve T cells need to undergo a transition from a spherical to a polarized shape prior to crawl over HEV. Therefore, the poor deformability manifested by *Arhgap45*^{-/-} naïve T cells might affect this prelude to TEM. Prior to entering LN, naïve T cells crawl along the HEV luminal surface in search of permissive extravasation sites. Force generation and deformability are both critical to allow squeezing through small openings in the endothelial barrier (Nourshargh et al., 2010). Disentangling whether the lack of ARHGAP45 affects the rolling to sticking transition and/or extravasation steps remains thus to be established via intravital video microscopy analysis.’

Consistent with the view that the rolling to sticking-transition is likely affected by the lack of ARHGAP45, in our microscopy observations of naïve T cells placed on 2D surface coated with ICAM-1 and chemokines (Figures 5 and 6 of our manuscript), we noted that at the onset of our sta recording, the percentage of cells that were immobile was significantly larger for naïve *Arghap45*^{-/-} T cells than for naïve WT T cells. It might result from the fact that after adhering on 2D-surface coated with ICAM-1 plus chemokine, naïve *Arghap45*^{-/-} T cells are less prone to lose their spherical shape and reach a polarized and motile state due to their poor deformability.

2. The normal T cell response in vivo in the draining LN suggests T cell motility in the LN may be less affected than suggested by the defective migration to chemokines in vitro. The authors

might comment about this. If sections showing transferred T cell distribution in WT spleen and LN are available, these could be informative.

As discussed above, we are not in a position to perform intravital video microscopy and address whether T cell motility in *Arhgap45*^{-/-} LN may be comparable to that of WT LN as suggested by the normal physiological T cell responses that are elicited in *Arhgap45*^{-/-} LN. A recent study (Hons et al., 2018) suggested that T cell intranodal migration occurs in a continuous sliding manner independently of prolonged cycles of adhesion and de-adhesion. Importantly, such mode of low-adhesive intranodal migration contrasts with the mode of migration invoked during T cell TEM, in which chemokines are thought to be upstream regulators of integrin activation for adhesion in a process known as inside-out signaling. This might explain the differential requirement of ARHGAP45 in T cell TEM and T cell intranodal migration. Accordingly, we added the following sentence in the revised Discussion:

'It has been suggested that T cell migration inside LN occurs in a continuous sliding manner independently of prolonged cycles of adhesion and de-adhesion (Hons et al., 2018). Importantly, such mode of low-adhesive intranodal migration differs from the mode of migration occurring during TEM, in which chemokines are thought to activate integrin adhesion in a process known as inside-out signaling. These different migratory modes might thus explain the differential requirement of ARHGAP45 in TEM and intranodal migration of T cells.'

3. Minor: it is stated that 'a similar trend (to LN) occurred in *Arhgap45* KO PP' but the data show a stronger phenotype.

We fully agree with Reviewer #2 and have corrected this point in the revised manuscript by changing the text to '*Arhgap45*^{-/-} LN had a reduced cellularity (Fig 2A) due to diminished numbers of naïve T (2.2-fold) and B (2.5-fold) cells (Fig 2D), whereas *Arhgap45*^{-/-} Peyer's patches showed a stronger reduction of naïve T (4-fold) and B (17-fold) cell numbers (Fig 2E)'.

Referee #2:

This manuscript provides a detailed and thorough analysis comparing T cells and B cells in and derived from wt versus arhgap45^{-/-} mice. In a separate and unlinked proteomic experiment, they describe the proteins associated with ARHGAP45 isolated from Jurkat T cells. While the data for the most part well controlled and reproducible, there is no information on the mechanistic basis for any of the analyses on T cells and B cells from the mice. Are all these phenotypes due to a change in RhoA activation or are they due to another function of ARHGAP45 or other interacting protein? There is no attempt to relate any of the interesting but preliminary proteomic analysis to the descriptive phenotypic analysis. Without any indication of the mechanism this has limited interest to scientists working on either Rho GTPase signaling or T/B cell signaling/function.

Points to consider for revisions:

1. English needs improving (grammar and choice of some words).

As suggested by Reviewer #2, we asked a colleague born in England to edit our revised manuscript to improve our English.

2. Abstract: 20 RhoGAPs (need to be specific because of Ras, Rab etc. GAPs).

As requested by Reviewer #2, we specified 'Rho GAPs' in the revised manuscript to be more specific.

3. Abstract: 'non-competitive setting/competitive setting' needs rewording - these are not familiar phrases for a non-specialist reader.

As requested by Reviewer #2, we reworded the corresponding sentence as follows: Under physiological conditions, ARHGAP45 controls TEM of naïve T and B cells whereas under competitive conditions it further regulates hematopoietic progenitor engraftment in the bone marrow and T-cell progenitor thymus-seeding.'

4. Introduction: The second paragraph needs references added throughout - there are none. There are 20 Rho GTPases in mammals, which are subdivided into 8 subgroups. GEF stands for guanine nucleotide exchange factor.

We fixed the omitted word 'nucleotide' in GEF and as requested by Reviewer #2 we also added the following references:

(Lawson and Ridley, 2018)

(Schulz et al., 2016)

5. Proposed RhoA specificity of ARHGAP45 - what about the highly homologous RhoB and RhoC? RhoC levels could be lower than RhoA because ARHGAP45 is a better GAP for RhoC than RhoA, hence RhoC is not bound efficiently because the GTP has been hydrolysed and it has dissociated from ARHGAP45.

As correctly pointed out by Reviewer #2, the 31-fold lower ARHGAP45-RhoC interaction stoichiometry as compared to the ARHGAP45-RhoA interaction stoichiometry can be due to the fact that 'ARHGAP45 is a better GAP for RhoC than RhoA, in that RhoC is not bound efficiently because the GTP has been hydrolysed and it has dissociated from ARHGAP45'. Accordingly, we added the following sentence in our revised manuscript to make it clear that RhoC levels could be lower than RhoA in our interactomics data because ARHGAP45 is a better GAP for RhoC than RhoA :

'The lower ARHGAP45-RHO C interaction stoichiometry might reflect that ARHGAP45 is a better GAP for RHO C than RHO A, leading to its rapid dissociation from ARHGAP45 after hydrolyzing the bound GTP.'

6. YWHAQ is 14-3-3 θ : it is important to state this when YWHAQ is first mentioned.

As requested by Reviewer #2 we have mentioned in the revised manuscript that YWHAQ is indeed 14-3-3 θ .

7. The mass spectrometry approach recognizes proteins that interact directly with ARHGAP45

and proteins that are associated with it indirectly via one or more intermediary proteins. It is not possible to distinguish between these two possibilities so 'interact' should be changed to 'associate' throughout the section on the 'interactome' (p. 5/6).

We fully agree with Reviewer #2 and changed 'interact' to 'associate' throughout the interactome paragraph and text.

8. For all graphs, absolute p values should be provided, not n.s. or asterisks. This allows readers to judge for themselves whether two values should be considered different or not. If numbers analysed vary between conditions in one experiment, then the numbers should be written above the condition directly, not given as a range in the figure legend. It is then much clearer which experimental conditions had fewer cells/mice.

We have checked the most recent issue of EMBO Reports (<https://www.embopress.org/journal/14693178>) and all the graphs shown in all the published papers showed p values that are provided exactly as we did in our original manuscript. We also contacted EMBO Reports Editorial Office and had the confirmation that the way we presented our data do fit the EMBO Reports 'Author guidelines'. Note that our Figures 2 and 8 are already extremely dense and following the suggestion of Reviewer #2 will have make them even more complicated. However, we followed Reviewer #2 instruction for the 'bigger' histograms found in Figure 3B, 3D and 4 (see point 9).

9. Figure 2: it is preferable to show the results for each mouse as a dot (as in Figure 3) and then the mean and SD on top of this. A good way to present this is to use different colors of dots for mice from each of the experiments, so that the reader can see if there is variation in the absolute numbers between different days or mice.

As requested by Reviewer #2 we showed in the revised Figure 2 the results for each mouse as a dot (as we did in all the other related figures) plus added the mean and SD on top of this. Although the 'use of different colors of dots for mice from each of the experiments' is not specified in the EMBO Reports 'Author guidelines' we did it for the 'bigger' histograms

found in Figure 3B, 3D, and 4. Due to their larger numbers and thus smaller individual size, attempting to do it in Figures 2 and 8 resulted in a total mess.

10. Figure 3B: It is not possible to compare results from 3 different experiments with *Arhgap45*^{-/-} mice with controls from only one experiment. The *Arhgap45*^{-/-} mice analysed on the same day as the control mice should be shown. The other mice from different days could be shown as separate bars for each day to indicate what the values were but not compared statistically.

We revised Figure 3B according to Reviewer #2 point 9 and the corresponding revised legend read: 'Data shown for *Arhgap45*^{-/-} mice correspond to 4 experiments (E1 to E4) involving a total of 15 individual mice whereas data shown for WT mice correspond to 2 experiments (E1 and E2) involving a total of 5 individual mice. The *Arhgap45*^{-/-} mice analyzed on the same day as the control WT mice corresponded to the E1 and E2 experiments'.

Note that WT-CMTPX/WT CTV are irrelevant to establish the key conclusion that ARHGAP45 deficiency impedes entry rate of naïve T and B cells in LN but not in spleen. Accordingly, to emphasize the histograms that need to be compared to reach our conclusion, we have reorganized the histograms to have side-by-side the WT/ *Arhgap45*^{-/-} B cell ratio in spleen and LN and the WT/ *Arhgap45*^{-/-} T cell ratio in spleen and LN. We further recalculated the p-value for the corresponding pairs of histograms.

11. Figure 3D and Figure 4: the results from all the mice analyzed should be shown here - it is not clear what 'representative' means. Why not show all the results - if they vary between different experiments this can be illustrated by using dots of different colors for each experiment?

As suggested by Reviewer #3, we revised Figure 3D and Figure 4 and the corresponding legends according to queries 10 and 11

12. Figure 5: As well as showing the mean speed of all cells from each experiment, it is essential to show the range of all cell speeds for each experiment with one dot for each cell, indicating the median cell speed. This is a standard way to represent cell speeds. It could be that mean

speed for all cells is not different, but the range of cell speeds varies more under some conditions (as suggested in Figure 6).

As suggested by Reviewer #2, we have added two panels in Figure 5 (panels B and F) to show individual cell speed. These plots are indeed helpful to exclude the possibility that distribution differences may have been hidden, and they support our conclusion that cell speed is similar for WT and *Arghap45*^{-/-} T activated T cells and different for WT and *Arghap45*^{-/-} T naïve T cells.

In the movies a proportion of cells are almost stationary, and this number could vary e.g. it could be higher for *arhgap45*^{-/-} cells or could vary between individual experiments. Although the authors claim there is no difference between wt and *arhgap45*^{-/-} activated T cells, the cells in the movies look different in their behavior and polarity. Notably, there are far fewer cells in the movies with *arhgap45*^{-/-} T cells than control cells (both activated and naïve). Is this because fewer *arhgap45*^{-/-} cells adhere under static conditions to ICAM1 or because they are less viable? In either case, quantification is required. (The flow assays do not provide the relevant information on adhesion under static conditions).

We have prepared a figure (see below) for the attention of Reviewer #2 where we plotted for each performed experiments the numbers of T cells that remained adherent after applying for 120 seconds a shear stress of 1 dyne/cm² (remaining cells were counted using a 5x magnification). They correspond to the number of cells shown in the movies at the onset of static condition video recording and they are on average 1.5-fold smaller for both naïve and activated *Arghap45*^{-/-} T cells than for their WT counterparts. As shown in Figure 5 (panels D and H), after applying for 120 seconds a shear stress of 1 dynes/cm², the percentage of naïve and activated *Arghap45*^{-/-} T cells remaining adherent was comparable to that of their naïve or activated WT counterparts. In addition, the naïve and activated WT and *Arghap45*^{-/-} T cells introduced into the imaging device were both > 97% viable as determined by Trypan blue exclusion. Therefore, the < 1.5 variations noted by Reviewer #2 are commensurable to the concentration of T cells we introduced into the imaging device. Note that in each experiment, we had enough cells to end up with good statistics without overloading our automatic cell tracking system.

13. Movie legends should state how many frames in the movie and time between each frame.

As requested by Reviewer #2, this issue has been fixed in the revised manuscript.

14. Transwell migration assays. It is surprising that 5 μm pore size was used - for T and B cells it is standard to use 3 μm pore size because of their small size compared to other cell types. The authors should repeat at least some of the experiments with 3 μm pores.

We respectfully disagree with comment 14 of Reviewer #2. The majority of studies involving Transwell migration assays conducted on T and B cells used 5 μm pore size. A rapid sampling of published papers (see below) involving leaders in the T/B cell chemotaxis field and including papers in highly regarded journals testifies that it is standard to use 5 μm pore size and not 3 μm pore size.

(Mohan et al., 2003, Laufer et al., 2018, Ueno et al., 2002, Beck et al., 2014, Estin et al., 2017, Wendland et al., 2011, Cordeiro Gomes et al., 2016, Hara-Chikuma et al., 2012, Moalli et al., 2018, Svensson et al., 2012, Zhang et al., 2019, Faroudi et al., 2010, Ottoson et al., 2001, Arnon et al., 2011)

Moreover, each panel of Figure 4 includes a 'no chemokine' control that shows that the T and B cells in the upper chamber do not passively 'fall' in the lower chamber. Furthermore, our Figure 4D confirmed that our assay measured chemotaxis (Fig 4D).

References

- ARNON, T. I., XU, Y., LO, C., PHAM, T., AN, J., COUGHLIN, S., DORN, G. W. & CYSTER, J. G. 2011. GRK2-dependent S1PR1 desensitization is required for lymphocytes to overcome their attraction to blood. *Science*, 333, 1898-903.
- BECK, T. C., GOMES, A. C., CYSTER, J. G. & PEREIRA, J. P. 2014. CXCR4 and a cell-extrinsic mechanism control immature B lymphocyte egress from bone marrow. *J Exp Med*, 211, 2567-81.
- CORDEIRO GOMES, A., HARA, T., LIM, V. Y., HERNDLER-BRANDSTETTER, D., NEVIUS, E., SUGIYAMA, T., TANI-ICHI, S., SCHLENNER, S., RICHIE, E., RODEWALD, H. R., FLAVELL, R. A., NAGASAWA, T., IKUTA, K. & PEREIRA, J. P. 2016. Hematopoietic Stem Cell Niches Produce Lineage-Instructive Signals to Control Multipotent Progenitor Differentiation. *Immunity*, 45, 1219-1231.
- ESTIN, M. L., THOMPSON, S. B., TRAXINGER, B., FISHER, M. H., FRIEDMAN, R. S. & JACOBELLI, J. 2017. Ena/VASP proteins regulate activated T-cell trafficking by promoting diapedesis during transendothelial migration. *Proc Natl Acad Sci U S A*, 114, E2901-E2910.
- FAROUDI, M., HONS, M., ZACHACZ, A., DUMONT, C., LYCK, R., STEIN, J. V. & TYBULEWICZ, V. L. 2010. Critical roles for Rac GTPases in T-cell migration to and within lymph nodes. *Blood*, 116, 5536-47.
- HARA-CHIKUMA, M., CHIKUMA, S., SUGIYAMA, Y., KABASHIMA, K., VERKMAN, A. S., INOUE, S. & MIYACHI, Y. 2012. Chemokine-dependent T cell migration requires aquaporin-3-mediated hydrogen peroxide uptake. *J Exp Med*, 209, 1743-52.
- HONS, M., KOPF, A., HAUSCHILD, R., LEITHNER, A., GAERTNER, F., ABE, J., RENKAWITZ, J., STEIN, J. V. & SIXT, M. 2018. Chemokines and integrins independently tune actin flow and substrate friction during intranodal migration of T cells. *Nat Immunol*, 19, 606-616.
- LAUFER, J. M., KINDINGER, I., ARTINGER, M., PAULI, A. & LEGLER, D. F. 2018. CCR7 Is Recruited to the Immunological Synapse, Acts as Co-stimulatory Molecule and Drives LFA-1 Clustering for Efficient T Cell Adhesion Through ZAP70. *Front Immunol*, 9, 3115.
- LAWSON, C. D. & RIDLEY, A. J. 2018. Rho GTPase signaling complexes in cell migration and invasion. *J Cell Biol*, 217, 447-457.
- MOALLI, F., FICHT, X., GERMAN, P., VLADYMYROV, M., STOLP, B., DE VRIES, I., LYCK, R., BALMER, J., FIOCCHI, A., KREUTZFELDT, M., MERKLER, D., IANNAcone, M., ARIGA, A., STOFFEL, M. H., SHARPE, J., BAhLER, M., SIXT, M., DIZ-MUNOZ, A. & STEIN, J. V. 2018. The Rho regulator Myosin IXb enables nonlymphoid tissue seeding of protective CD8(+) T cells. *J Exp Med*, 215, 1869-1890.
- MOHAN, K., PINTO, D. & ISSEKUTZ, T. B. 2003. Identification of tissue transglutaminase as a novel molecule involved in human CD8+ T cell transendothelial migration. *J Immunol*, 171, 3179-86.
- NOURSHARGH, S., HORDIJK, P. L. & SIXT, M. 2010. Breaching multiple barriers: leukocyte motility through venular walls and the interstitium. *Nat Rev Mol Cell Biol*, 11, 366-78.
- OTTOSON, N. C., PRIBILA, J. T., CHAN, A. S. & SHIMIZU, Y. 2001. Cutting edge: T cell migration regulated by CXCR4 chemokine receptor signaling to ZAP-70 tyrosine kinase. *J Immunol*, 167, 1857-61.

- SCHULZ, O., HAMMERSCHMIDT, S. I., MOSCHOVAKIS, G. L. & FORSTER, R. 2016. Chemokines and Chemokine Receptors in Lymphoid Tissue Dynamics. *Annu Rev Immunol*, 34, 203-42.
- STEIN, J. V. & RUEF, N. 2019. Regulation of global CD8(+) T-cell positioning by the actomyosin cytoskeleton. *Immunol Rev*, 289, 232-249.
- SVENSSON, L., STANLEY, P., WILLENBROCK, F. & HOGG, N. 2012. The Galphaq/11 proteins contribute to T lymphocyte migration by promoting turnover of integrin LFA-1 through recycling. *PLoS One*, 7, e38517.
- UENO, T., HARA, K., WILLIS, M. S., MALIN, M. A., HOPKEN, U. E., GRAY, D. H., MATSUSHIMA, K., LIPP, M., SPRINGER, T. A., BOYD, R. L., YOSHIE, O. & TAKAHAMA, Y. 2002. Role for CCR7 ligands in the emigration of newly generated T lymphocytes from the neonatal thymus. *Immunity*, 16, 205-18.
- WENDLAND, M., WILLENZON, S., KOCKS, J., DAVALOS-MISSLITZ, A. C., HAMMERSCHMIDT, S. I., SCHUMANN, K., KREMMER, E., SIXT, M., HOFFMEYER, A., PABST, O. & FORSTER, R. 2011. Lymph node T cell homeostasis relies on steady state homing of dendritic cells. *Immunity*, 35, 945-57.
- ZHANG, Q., HE, Y., LUO, N., PATEL, S. J., HAN, Y., GAO, R., MODAK, M., CAROTTA, S., HASLINGER, C., KIND, D., PEET, G. W., ZHONG, G., LU, S., ZHU, W., MAO, Y., XIAO, M., BERGMANN, M., HU, X., KERKAR, S. P., VOGT, A. B., PFLANZ, S., LIU, K., PENG, J., REN, X. & ZHANG, Z. 2019. Landscape and Dynamics of Single Immune Cells in Hepatocellular Carcinoma. *Cell*, 179, 829-845 e20.

Dear Dr. Malissen,

Thank you for the submission of your revised manuscript to our editorial offices. I have now received the report from the two referees that were asked to re-evaluate your study, you will find below. As you will see, the referees now support the publication of your study in EMBO reports, but they both have some final comments/further suggestions to improve the manuscript, I ask you to address in a final revised version. Please also provide a point-by-point-response addressing the remaining points by the referee.

Regarding the major remaining point of referee #2, I think the proteomics data should stay. However, please clearly discuss the limitations of the ARHGAP45 interactome, and moderate the conclusions as indicated by the referee ('The statement: 'Therefore, the composition of the ARHGAP45 interactome of Jurkat T cells demonstrates that ARHGAP45 acts as a GAP specific for RHO and presumably RHOC' is highly speculative and not supported by any other results.').

Further, I have these editorial requests:

- Please have you final manuscript text carefully proofread by a native speaker. There are still several grammatical errors or typos present. See also the report of referee #2.
- Figure EV4C is also shown in panel 6C. Please explain/mention this in the respective figure legends.
- Please remove the statement 'Expanded View for this article is available on line' from the manuscript text.
- It seems Hui Wang is missing from the author contributions. Please check. Moreover, HL should be corrected to LH, and FH should be correct to FHZ. Please check again that all authors are mentioned correctly in this section.
- Please format the references exactly according to our new reference format:
<http://www.embopress.org/page/journal/14693178/authorguide#referencesformat>
- Please make sure that all the funding information is entered into the online submission system, and is complete and similar to the one in the manuscript text file.
- Please add separate callouts for Figs EV2-EV5 to the manuscript text.
- The Dataset table callouts need correcting to Dataset EV1 and Dataset EV2 (not 'supplementary' nor 'excel').
- Please remove the legends for the datasets from the manuscript file. Please add these on the first TAB of the respective excel files.
- Please also remove the legends for the movies from the manuscript text. Please provide these as separate text files ZIPed together with the movie files and uploaded ZIPed together.
- Please name the headline for the figure legends for the EV figures 'Expanded view figure legends'.

- As they are significantly cropped, could you provide the source data for the few Western blots shown in the manuscript (including the EV figures)? The source data will be published in separate source data files online along with the accepted manuscript and will be linked to the relevant figures. Please submit scans of entire gels or blots together with the revised manuscript. Please include size markers for scans of entire gels, label the scans with figure and panel number, and send one PDF file per figure.

- Finally, please find attached a word file of the manuscript text (provided by our publisher) with changes we ask you to include in your final manuscript text, and some queries, we ask you to address. Please provide your final manuscript file with track changes, in order that we can see any modifications done.

In addition I would need from you:

- a short, two-sentence summary of the manuscript
- two to three bullet points highlighting the key findings of your study
- a schematic summary figure (in jpeg or tiff format with the exact width of 550 pixels and a height of not more than 400 pixels) that can be used as a visual synopsis on our website.

Kind regards,

Achim

Achim Breiling
Editor
EMBO Reports

Referee #1:

In their response to reviewers and revisions to the manuscript, the authors have adequately addressed my concerns. In the revised manuscript (based on data shared with reviewers) the authors now indicate that cell attachment to endothelium could be affected perhaps as much as the transmigration step. With that in mind, I suggest changing the title e.g. "ARHGAP45 controls naive T and B cell entry into lymph nodes and T cell progenitor thymus-seeding"

Referee #2:

Comments on revisions:

Given that the authors have not addressed my comments on the lack of mechanistic understanding of the phenotypes observed with T/B cells, and that the proteomic data are still preliminary, with no revisions, I strongly recommend that the proteomic data are removed. There is no verification that any of the ARHGAP45-associated proteins identified here actually co-immunoprecipitate with ARHGAP45, or that they have any functional link to ARHGAP45. The

statement: 'Therefore, the composition of the ARHGAP45 interactome of Jurkat T cells demonstrates that ARHGAP45 acts as a GAP specific for RHO and presumably RHOC' is highly speculative and not supported by any other results.

1. English needs improving (grammar and choice of some words):

This are still some errors, particularly for plural versus singular, correct article ('the/a' is missing or incorrect) e.g. Introduction: RhoGAPs and RhoGEFs (plural whenever referring to more than one); GEFs = guanine nucleotide exchange factors; a RhoGAP domain (not domains); of a helical bundle; the actin cytoskeleton. Please also carefully check the entire text for similar problems (including the figure legends).

2. Introduction and all other occurrences: RhoGAPs not GAPs - you only changed this in the Abstract.

3. Abstract: You have still not reworded 'competitive setting' - this needs to be changed.

4. Introduction: There are 20 Rho GTPases in mammals, which are subdivided into 8 subfamilies NOT just 'Rho, Rac and Cdc42 subfamilies' - this is still not changed.

5. You state in your reply that 'the naive and activated WT and Arghap45^{-/-} T cells introduced into the imaging device were both > 97% viable as determined by Trypan blue exclusion'. Please add this information to the relevant methods section.

Point-by-point responses**Referee #1:**

In their response to reviewers and revisions to the manuscript, the authors have adequately addressed my concerns. In the revised manuscript (based on data shared with reviewers) the authors now indicate that cell attachment to endothelium could be affected perhaps as much as the transmigration step. With that in mind, I suggest changing the title e.g. "ARHGAP45 controls naive T and B cell entry into lymph nodes and T cell progenitor thymus-seeding".

It is an excellent suggestion and we have thus used the title suggested by Referee #1.

Referee #2:

Given that the authors have not addressed my comments on the lack of mechanistic understanding of the phenotypes observed with T/B cells, and that the proteomic data are still preliminary, with no revisions, I strongly recommend that the proteomic data are removed. There is no verification that any of the ARHGAP45-associated proteins identified here actually co-immunoprecipitate with ARHGAP45, or that they have any functional link to ARHGAP45. The statement: 'Therefore, the composition of the ARHGAP45 interactome of Jurkat T cells demonstrates that ARHGAP45 acts as a GAP specific for RHO and presumably RHOC' is highly speculative and not supported by any other results.

We mitigated our conclusion and substituted 'demonstrates' by 'suggests':

Therefore, the composition of the ARHGAP45 interactome of Jurkat T cells suggests that ARHGAP45 acts as a GAP specific for RHO and presumably RHOC.

Moreover, we removed the claim that ARHGAP45 is RHOA GAP in the Abstract.

1. English needs improving (grammar and choice of some words): This are still some errors, particularly for plural versus singular, correct article ('the/a' is missing or incorrect) e.g. Introduction: RhoGAPs and RhoGEFs (plural whenever referring to more than one); GEFs = guanine nucleotide exchange factors; a RhoGAP domain (not

domains); of a helical bundle; the actin cytoskeleton. Please also carefully check the entire text for similar problems (including the figure legends).

Fixed

2. Introduction and all other occurrences: RhoGAPs not GAPs - you only changed this in the Abstract.

Fixed

3. Abstract: You have still not reworded 'competitive setting' - this needs to be changed.

Done : Under physiological conditions, ARHGAP45 controls the entry of naïve T and B cells into lymph nodes whereas under competitive repopulation it further regulates hematopoietic progenitor cell engraftment in the bone marrow and T-cell progenitor thymus-seeding.

Please note that 'competitive repopulation' is routinely used in Immunology and Hematology

4. Introduction: There are 20 Rho GTPases in mammals, which are subdivided into 8 subfamilies NOT just 'Rho, Rac and Cdc42 subfamilies' - this is still not changed.

Fixed: Among the eight sub-families that constitute the RHO family of small GTPases, the RAC, RHO and CDC42 sub-families control cell polarity, shape, and migration by regulating actin cytoskeletal dynamics.

5. You state in your reply that 'the naive and activated WT and Arghap45^{-/-} T cells introduced into the imaging device were both > 97% viable as determined by Trypan blue exclusion'. Please add this information to the relevant methods section.

This information has been added in the paragraph 'Migration assays' : Control and ARGAP45-KO naïve mouse T cells (5 x10⁴ cells/mL and > 97% viable as determined by Trypan blue exclusion) were plated in channels and placed in a humidified incubator at 37°C and 5% CO₂ for 10 min before observation.

Bernard Malissen
CNRS
Centre d'Immunologie INSERM
CNRS de Marseille-Luminy
Case 906
Marseille Cedex 9, F-13288 Marseille Cedex 9 13288
France

Dear Dr. Malissen,

I am very pleased to accept your manuscript for publication in the next available issue of EMBO reports. Thank you for your contribution to our journal. Please make sure the mass spectrometry proteomics data deposited at the ProteomeXchange Consortium via the PRIDE partner repository is public upon publication of the paper.

At the end of this email I include important information about how to proceed. Please ensure that you take the time to read the information and complete and return the necessary forms to allow us to publish your manuscript as quickly as possible.

As part of the EMBO publication's Transparent Editorial Process, EMBO reports publishes online a Review Process File to accompany accepted manuscripts. As you are aware, this File will be published in conjunction with your paper and will include the referee reports, your point-by-point response and all pertinent correspondence relating to the manuscript.

If you do NOT want this File to be published, please inform the editorial office within 2 days, if you have not done so already, otherwise the File will be published by default [contact: emboreports@embo.org]. If you do opt out, the Review Process File link will point to the following statement: "No Review Process File is available with this article, as the authors have chosen not to make the review process public in this case."

Should you be planning a Press Release on your article, please get in contact with emboreports@wiley.com as early as possible, in order to coordinate publication and release dates.

Thank you again for your contribution to EMBO reports and congratulations on a successful publication. Please consider us again in the future for your most exciting work.

Yours sincerely,

Achim Breiling
Editor
EMBO Reports

THINGS TO DO NOW:

You will receive proofs by e-mail approximately 2-3 weeks after all relevant files have been sent to our Production Office; you should return your corrections within 2 days of receiving the proofs.

Please inform us if there is likely to be any difficulty in reaching you at the above address at that time. Failure to meet our deadlines may result in a delay of publication, or publication without your corrections.

All further communications concerning your paper should quote reference number EMBOR-2020-52196V3 and be addressed to emboreports@wiley.com.

Should you be planning a Press Release on your article, please get in contact with emboreports@wiley.com as early as possible, in order to coordinate publication and release dates.

Corresponding Author Name: Bernard MALISSEN

Journal Submitted to: EMBO REPORTS

EMBOR-2020-52196V3